# Effective Unsupervised Subset Selection Through The Lens of Neural Network Pruning

## Abstract

Having large amounts of annotated data significantly impacts the effectiveness of deep neural networks. However, the annotation task can be very expensive in some domains, such as medical data. Thus, it is important to select the data to be annotated wisely, which is known as the subset selection problem. We investigate and establish a relationship between subset selection and neural network pruning, which is more widely studied. Leveraging insights from network pruning, we propose utilizing the norm criterion of neural network features to improve subset selection methods. We empirically validate our proposed strategy on various networks and datasets, demonstrating enhanced accuracy. This shows the potential of employing pruning tools for subset selection.

## 1 Introduction

An important factor in the success of deep neural networks is having a large set of annotated data. However, it can be difficult to obtain such labeled data. For example, in medical applications, annotation must be done by expert doctors whose time is expensive. Consequently, given limited resources, it is imperative to select the examples to be annotated from a large unlabeled dataset wisely, so as to extract maximum information from the data. Our work focuses on this challenge, known as the *subset selection problem.*

The subset selection problem is closely related to active learning, which involves gradually selecting unlabeled examples for annotation during the learning process. While gradual annotation has benefits, subset selection is the special case where all the samples to be annotated must be selected at once, all together. Subset selection poses several challenges. Firstly, determining the criteria for selecting informative examples is non-trivial, as it requires balancing various factors, such as diversity, relevance, and coverage of the data distribution. Subset selection based on simple criteria such as uncertainty, entropy, and margin between the highest scores has been demonstrated to be ineffective. Moreover, it has been shown that many methods proposed for subset selection fail to outperform random choice, particularly when a very small set of examples is chosen (Hacohen et al., 2022; Chen et al., 2022; Guo et al., 2022).

In this study, we explore the relationship between training-data subset selection and neural network pruning, which is a well-researched area (Mozer & Smolensky, 1988; Molchanov et al., 2016; LeCun et al., 1989; Hassibi et al., 1993; Chauvin, 1988; Carreira-Perpiñán & Idelbayev, 2018; Louizos et al., 2018; Bellec et al., 2018; Mocanu et al., 2018; Mostafa & Wang, 2019; Novikov et al., 2015; Jaderberg et al., 2014; Chen et al., 2023; 2021; Neill, 2020; van Amersfoort et al., 2020; Lee et al., 2018; de Jorge et al., 2020; Alizadeh et al., 2022; Paul et al., 2022; Lee et al., 2020b; Wang et al., 2020; Zhang et al., 2021b; Su et al., 2020; Liu & Zenke, 2020; Sun et al., 2024). Neural pruning reduces the computational costs of training and inference in deep models (see Vadera & Ameen (2022) for a survey).

We propose that input data can be seen as part of the neural network structure, suggesting that methods for pruning weights can also be applied to 'prune' the training examples (see Fig. 1). Motivated by this relation, we focus on migrating to subset selection the use of norms of network features that is practiced in pruning. Structured pruning methods have shown the effectiveness of retaining filters with high norms while discarding those with low norms (Li et al., 2017; He et al., 2018; 2019). Similarly, unstructured pruning techniques, particularly at initialization, prioritize weights with higher magnitudes (Frankle & Carbin, 2018).

Building upon these insights from prior works, we propose leveraging the norm criterion of features in the data to enhance the efficacy of selection methods. We start by investigating how the norm of the features is a contributor to subset performance by using the norms' values as a probability from which the subset is sampled. Our findings reveal that subsets characterized by high norms exhibit superior accuracy following training. Therefore, we suggest a weighted sampling criterion that relies on the norm. Additionally, we tested the accuracy of numerous subsets sampled uniformly at random and observed a correlation between norm and accuracy, indicating the significance of norm in determining the effectiveness of the selected subsets.

Relying solely on examples' feature norms is limited, as it misses data correlations. Thus, we suggest using the Gram-Schmidt process to stably choose examples orthogonal to those already selected. This facilitates selecting examples whose features have the highest norm in the 'remaining subspace' that is not spanned by the previously selected examples. Handling the data correlations in this way promotes a comprehensive coverage of the features' domain.

We motivate our proposed approach through the 'classic' EigenFaces algorithm (Sirovich & Kirby, 1987; Turk & Pentland, 1991), which relies on projections to a linear feature space for the task of face recognition. Intuitively, we aim to select examples that retain the most energy in this space, as they are the most informative for the classification process. This intuition provides a basis for our norm-based selection criterion. However, selecting only the examples with the largest norms may result in a lack of diversity among the chosen samples. To address this, we introduce randomization, weighting the selection process by the norm to ensure diversity and select examples that are less correlated. These natural design choices motivate our use of randomization in conjunction with the norm-based criterion and the Gram-Schmidt algorithm. We demonstrate empirically that indeed, selecting subsets in an unsupervised manner using the norm-based criterion and the Gram-Schmidt process effectively identifies informative subsets in the linear case.

Similar to the linear case, where PCA is used, in the non-linear case, we aim at maximizing the energy kept in the feature space created by a neural network. We present results with features induced by fully random networks without training and self-supervised networks, namely, SimCLR (Chen et al., 2020) and DINO (Caron et al., 2021), on the CIFAR-10/100, Tiny-ImageNet, ImageNet and OrganAMNIST (Yang et al., 2023) datasets. We show that combining our norm criterion with other subset selection methods can significantly improve the overall performance, and thus achieve new state-of-the-art results in most cases. We demonstrate that our approach is versatile, performs well across frameworks and is applicable to various feature domains.

Our contributions are summarized as follows: (i) A novel relation between pruning and subset selection; (ii) Proposing the network's feature norm and Gram-Schmidt algorithm as successful subset selection tools; and (iii) Comprehensive experiments demonstrating our framework's advantages.

## 2  Related Work

Our work focuses on the unsupervised subset selection problem. While both subset selection and active learning share the goal of selecting informative examples for labeling, the main difference lies in their approach and methodology. Subset selection focuses on choosing a fixed subset of examples from the entire dataset, while active learning involves dynamically selecting examples for labeling based on the current model's uncertainty or informativeness. Unsupervised subset selection is particularly useful when labeling is costly and must be performed in a single batch. In contrast, active learning requires an annotator to be available for multiple iterations of the classification process. Notably, in subset selection, the chosen subsets should ideally be model-agnostic and capable of achieving good performance when training the model from scratch. Some of the active learning methods can be used without an initial labeled set and do not heavily rely on the labels chosen in previous iterations and thus can be used as a subset selection with a single iteration.

**Subset Selection** strategies for choosing examples include uncertainty sampling (Gal et al., 2017; Settles, 2009), diversity maximization (Ash et al., 2019; Citovsky et al., 2021; Beluch et al., 2018; Hsu & Lin, 2015; Chang et al., 2017; Chen et al., 2020), and entropy criteria (Coleman et al., 2019). Generally, these approaches aim to select examples that best represent the underlying data distribution while minimizing redundancy. In active learning framework, numerous methods are tailored to the geometry of the problem and often rely on k-means clustering to select diverse sets (Hacohen et al., 2022; Sener & Savarese, 2018; Sorscher et al.,

(a) **Structured pruning:** Selection of a small subset of neurons. (b) **Subset Selection:** Choosing a small subset of training examples.

Figure 1: Subset selection is analogous to pruning of the input data, which can be considered the first layer of the network. Gray parameter rows in Fig. 1a represent removed neurons and gray training data rows in Fig. 1b represent removed data points. In our work, we do not choose examples directly from their raw input but rather use an unsupervised feature extractor.

2022; Xia et al., 2022). Others aim to cover the embedding space (Yehuda et al., 2022; Zheng et al., 2022a). Others adopt an optimization perspective (Borsos et al., 2020; Paul et al., 2021; Killamsetty et al., 2021b), suggesting the estimation of the gradient of the entire dataset using a small amount of data (Killamsetty et al., 2021a; Mirzasoleiman et al., 2020).

Recent challenges in subset selection focus on choosing examples from a labeled pool rather than an unlabeled one (Gadre et al., 2024; Mazumder et al., 2024). Moreover, Guo et al. (2022) introduce a framework for subset selection.

The works relevant to our research are the ProbCover (Yehuda et al., 2022) and TypiClust (Hacohen et al., 2022) algorithms, which to the best of our knowledge are the current state-of-the-art extremely small subset selection approaches for the image classification task. ProbCover is a novel subset selection strategy designed to address the challenges of deep active learning in low-budget regimes. It leverages recent advancements in self-supervised learning to enrich the geometry of data representations, enabling more effective subset selection for annotation. By maximizing Probability Coverage, ProbCover aims to select examples that contribute the most information to the learning process, thereby reducing annotation costs while least harming performance. TypiClust is a subset selection technique specifically tailored to low-budget scenarios, where only a limited number of labeled examples are available for training. Based on theoretical analysis revealing a phase transition-like behavior, TypiClust employs a querying strategy that prioritizes typical examples when the budget is constrained and unrepresentative examples when the budget is larger. This approach capitalizes on the observed phenomenon that typical examples contribute the most information in low-budget settings, leading to improved model performance. Both of these method depend heavily on the feature domain, and their performance drops when the features are uninformative. Moreover, methods such as ProbCover and TypiClust require the computation of clustering or adjacency graphs. In contrast, we demonstrate that our method does not rely heavily on the domain and is less time-consuming.

**Neural Network Pruning** has emerged as a technique for reducing model complexity and compute, see (Vadera & Ameen, 2022) for a comprehensive survey. Various pruning methods have proposed using norm-based techniques, graduality, and randomization. Norm or magnitude based techniques discard parameters with low magnitudes, often achieving significant compression with only a minor harm to performance (Han et al., 2015; He et al., 2019; 2018; Li et al., 2016; Frankle & Carbin, 2018; See et al., 2016; Guo et al., 2016; Narang et al., 2016; Tung & Mori, 2018; Lubana & Dick, 2020; Sun et al., 2024). Gradual pruning iteratively removes the least important weights, allowing networks to adapt gradually and maintain performance (Frankle & Carbin, 2018; Han et al., 2015; Lee et al., 2020a). Finally, using randomization in the pruning process further enhances performance (Bar & Giryes, 2023; He et al., 2019).

We focus on interpreting subset selection via pruning, utilizing the features' norm and randomization to subset selection together with an algorithm to ensure feature diversity.

## 3 Pruning and Subset Selection

**Problem Statement.** Given an unlabeled large dataset with $D = \{x_1, ..., x_N\}$ the question is to select a small subset of $s \ll N$ examples. The selected exampled are labeled and we get a small labeled subset, $S = \{(x_1, y_1), ...(x_s, y_s)\}$. The goal is to maximize the performance of the model after training, utilizing these small labeled subsets. In our work, we assume that each unlabeled example, $x_i$, has a correspondig feature vector $F_i$ which is usually derived from an unsupervised trained model, $F_i = \mathcal{M}(x_i; \theta)$.

**Subset Selection as Pruning.** The key in our work is the analogy we draw between pruning and subset selection: selecting examples from the dataset is analogous to choosing filters in the pruning setting. Specifically, viewing input examples as the very first layer of the network suggests that choosing a filter corresponds to selecting an example. Both, the pruned network weights and the input examples that are not in the selected subset are treated as zero. Neither 'unselected' examples nor pruned weights participate in the training of the network. Fig. 1 provides a visual illustration. Indeed, the correspondence holds when filter pruning is performed in a layer-wise manner and all other layers remain static; thus, this is simply an analogy and there are clear differences between the two cases.

The features' norm is the first tool we migrate from pruning. Extensive research in network pruning has emphasized the significance of high-norm filters for structured pruning (He et al., 2018; 2019), high-magnitude weights for unstructured pruning (Frankle & Carbin, 2018) and the norms of features (Li et al., 2017). In Appendix I we further show the bias of pruned network to high feature norms.

Motivated by norm-based pruning, we use the norm of features for subset selection. The features can be obtained through various methods, including pre-training procedures or randomly initialized networks. We focus on the norms of the features and not the input itself. Clearly, the input impacts the features created throughout the network layers, and hence relying on the features norm is data driven.

**Feature Norm.** Let $F_1, ..., F_N$ denote the features corresponding to $N$ unlabeled training examples. Note that these features are the output of the neural network prior to the last linear classification layer. Given these features, we randomize the examples according to the following probability:

$$p_i = \frac{\|F_i\|}{\sum_{j=1}^{N} \|F_j\|}, \quad i = 1, ..., N, \qquad (1)$$

where $\|\cdot\|$ is the $\ell_2$ norm unless otherwise stated. In Section 4, we demonstrate that this simple choice of examples performs better than uniform random selection, which is a non-trivial baseline, particularly with extremely small subsets (Hacohen et al., 2022; Chen et al., 2022). Moreover, we show that there is an advantage to randomization over deterministic selection of the highest weights.

---

**Algorithm 1** Gram–Schmidt for Subset Selection

**Input:** Feature extractor model $\mathcal{M}$, unlabeled examples $\{x_i\}_{i=1}^N$ and $s$ number of examples to label.

**Feature Extraction:** $F_i = \mathcal{M}(x_i)$
**Initialize:** $\tilde{F}_i = F_i$ and $S = \emptyset$
**for** $k = 1$ **to** $s$ **do**
    **Randomize** $i$ according to $p_j = \frac{\|\tilde{F}_j\|}{\sum_{k=1}^{N} \|\tilde{F}_k\|}, j \notin S$

    **Update:** $S = S \cup \{i\}$
    **Projection:** For $j \notin S$: $\tilde{F}_j = \tilde{F}_j - \frac{\tilde{F}_j^T \tilde{F}_i}{\|\tilde{F}_i\|^2} \tilde{F}_i$
**end for**
**return** $S$

---

**Motivation for features norm for selection.**
Given a meaningful feature extractor model $\mathcal{M}(x; \theta)$, the norm of the extracted features $\mathcal{M}(x_i; \theta)$ for an input $x_i$ can serve as an indicator of how well $x_i$ aligns with the feature space defined by $\mathcal{M}$. A higher norm suggests that $x_i$ exhibits stronger correlation with the feature extractor's learned representation. If the feature space is indeed meaningful for the task, such correlation implies that $\|x_i\|$ is informative.

**Gram-Schmidt.** In the context of subset selection, it is crucial to emphasize that solely relying on feature norms may not yield optimal results. While feature norms provide valuable information, they may not capture the full complexity of the dataset. Specifically, the norm values do not contain information about the correlations between data points. Therefore, it is essential to augment norm-based selection methods with additional concepts to enhance their effectiveness. We suggest leveraging techniques from linear algebra and choosing features that span the domain. Namely, we utilize the Gram-Schmidt process described in Algorithm 1 to iteratively choose orthogonal features. Initially, we derive the unsupervised features according to the unsupervised model $\mathcal{M}$. The first step of the selection is to randomly select an example to be labeled

according to the norms using the probabilities in Eq. (1). Then, we update the set of chosen examples, $S$. Finally, we remove the projection onto the chosen feature from the remaining features, which is the third step of the Gram-Schmidt process, ensuring orthogonality of the remaining features to the selected ones. In this way, we handle the correlations between the data points. We ensure that the chosen examples not only have high feature norms, but also capture diverse and informative aspects of the dataset.

The Gram-Schmidt process is known to be numerically unstable when the projections are calculated with the initial inputs rather than gradually. We address this issue in Algorithm 1 by performing the projection with $\tilde{F}_i$ rather than with the initial input vectors $F_i$, and update the values of the features, $\{\tilde{F}_j\}_{j \notin S}$, in each iteration.

The method we suggest enjoy low computation complexity. Let $d$ be the dimension of the features. In each iteration, the randomization step takes $O(Nd)$ to calculate the features' norms and normalize them to probabilities. For randomization according to the norm (Eq. (1)), the complexity of weights sampling of a vector of dimension $N$ is $O(\log N)$ and the sampling is performed $s$ times. Hence, the overall complexity of norm randomization is $O(Nd + s \log N)$. For Algorithm 1, the projection step takes $O(Nd)$ in the worst case for calculating the inner products of the $N - 1$ remaining features with the chosen feature. Overall complexity is $O(sNd)$. Of course, when $s = O(N)$ the complexity becomes $O(N^2 d)$. In comparison, the baseline, ProbCover, requires $O(N^2 d)$ computations. We assume that the features are given and we do not include their query to the complexity calculations since both ProbCover and our Gram-Schmidt rely on the availability of the features.

**Randomization.** In our approach, we incorporate randomization into the selection process. Prior work on pruning methods has shown that adding randomization to a saliency score can improve performance (Bar & Giryes, 2023). This can be viewed as sketching the model's output features. Building on this idea, we adopt a similar strategy: Instead of selecting examples with the highest norm values, which serve as our saliency scores, we perform a weighted random selection based on these scores. We argue that using randomization is crucial for achieving good performance. In Fig. 6c we demonstrate the role of randomization in subset selection performance. In the warm-up example below (Section 3.1), we provide another motivation for using randomization, which is ensuring diversity.

## 3.1 Warm-up Example

In this section, we demonstrate the benefits of our methods using the simple EigenFaces algorithm (Sirovich & Kirby, 1987; Turk & Pentland, 1991). This demonstration will provide also an extra motivation for our proposed approach. EigenFaces is a classical algorithm used for face recognition. In this problem, we have $x_1, .., x_n$ images of faces from $p$ different people and the person in each image is labeled. The labels are denoted by $y_1, ..., y_n$. The pre-processing step is simply calculating the PCA of the input: $W = \text{PCA}(x_1, .., x_n)$ and projecting the input to the main principal components, which hold the important information of the data. Thus, the PCA is used as an informative feature extractor. The projection of the input using $W$ results in the features: $\{Wx_i\}_{i=1}^n$, where we abuse notation here and denote by $W$ the projection onto the main components of the PCA. For classification, the model returns simply the class of the nearest neighbor after the PCA projection. Formally,

$$ j = \arg\min_{i \in [n]} \{\|Wx_i - Wx\|\}, \qquad \text{Pred}(x) = y_j. \tag{2} $$

In the subset selection task, $W$ is the unsupervised feature extractor and we may choose the subset according to the features, $\{Wx_i\}_{i=1}^n$, to form a small subset $S = \{i_1, ..., i_s\}$. We calculate the linear transformation, $W_s$, based on the selected examples. Following the selection step, the labels of the chosen small subset are utilized to classify the faces (Eq. (2)).

For this problem, selecting examples with high energy values (i.e., higher norms) after PCA projection is a natural choice as they are better aligned with the feature domain. Because PCA provides a simple yet informative representation of the feature space, high-energy examples tend to correspond to more informative data. Consequently, selecting a small set of high-energy examples for classification effectively minimizes performance degradation associated with using small subsets.

| Method | Accuracy | Coverage | Accuracy | Coverage |
|---|---|---|---|---|
| Subset size | 40 | | 80 | |
| Random | 43.13 | 27 | 65.00 | 35 |
| Max Norm | 30.00 | 16 | 45.63 | 23 |
| Norm Randomization | 49.38 | 26 | 71.25 | **37** |
| Gram-Schmidt | **58.13** | **32** | **75.00** | **37** |

Table 1: Warmup example: Face recognition task with EigenFaces. Comparison of accuracy and class coverage. The number of classes is 40.

It is important to note that relying solely on the norm criterion may result in selecting examples predominantly from a single class, particularly if that class has higher energy than others. To address this, ensuring diversity in the selection is crucial. This can be achieved by incorporating randomization into the selection process or by decorrelating the information already present in the selected examples, as is done with the Gram-Schmidt approach.

We asses our strategy with the EigenFaces algorithm and The AT&T Dataset of Faces. This dataset contains 400 images of faces of 40 people. We randomly split the data into 240 train and 160 test balanced sets. We apply our approaches with randomization according to norm (Eq. (1)) and Gram-Schmidt (Algorithm 1) and additionally compare them with uniform at random sampling and choose according to the maximal norm. We report accuracy and class coverage in the chosen subset.

The results in Table 1 show the benefits of our methods. Norm randomization induces good results and coverage. Norm without randomization suffers from a major degradation in coverage which is probably the cause of its failure. The accuracy and the coverage are highest with Gram-Schmidt supporting the hypothesis that a good representation of the domain is beneficial. Additionally, it is clear that the performance does not only rely on the coverage of the classes, but that the quality of the chosen examples is crucial too. This can be observed in the case of selecting 40 examples where random sampling has better coverage but norm randomization achieves better accuracy. Also, for 80 examples Gram-Schmidt enjoys better performance than norm randomization even with the same class coverage. Qualitative results of images selected using different methods are presented in Appendix K, which demonstrate that our approach selects a more diverse set of faces. Having motivated our proposed framework in the linear case, we move test it in the non-linear case.

## 4 Experiments

We validated our proposed strategy empirically across diverse settings and datasets, focusing on extremely small subsets selected from unlabeled pools. Consequently, in many instances, not all classes are represented in the labeled training set, leading to potential class imbalances within the subsets.

We evaluate our method in three frameworks: (i) *Fully supervised*: Training exclusively with annotated data using an initialized ResNet-18 from scratch. (ii) *Semi-supervised with linear probing*: Training a single-layer linear classifier on top of self-supervised features obtained from an unlabeled dataset. This framework aims to leverage semi-supervised learning principles without relying on recent advances in pseudo-labeling techniques. (iii) *Semi-supervised*: Training competitive semi-supervised methods using subsets chosen by the selection algorithms. We employ FlexMatch (Zhang et al., 2021a) and SimMatch (Zheng et al., 2022b) to assess the effectiveness of our selection algorithm.

We present results with randomization along the feature norms approach according to Eq. (1), the Gram-Schmidt based strategy (Algorithm 1), and a combination of ProbCover (Yehuda et al., 2022) and TypiClust (Hacohen et al., 2022) with the norm criterion. ProbCover and TypiClust rely on self-supervised embeddings. ProbCover is a method which selects examples that contribute the most information to the learning process and TypyClust prioritizes typical examples to improve the performance with extremely low budgets. We combined the methods with the norm criterion and GS (Algorithm 1), named "*<method> then norm*" or "*<method> then GS*", by selecting twice the required budget, denoted as $2b$, and then randomly selecting $b$

examples based on feature norms. We choose to embed the norm criterion this way since the methods rely on normalized features and are very sensitive to changes in the feature domain.

The same methodology of combining norms and Gram-Schmidt can be used for other existing methods when features are available. Selecting examples based on their features introduces only a minor computational overhead to these methods. As we demonstrate below, this added computation leads to performance improvement in the vast majority of cases. In the paper, we only show the combination of the feature norm criterion with these methods. In Appendix E, we also show results with Gram-Schmidt on top of the baselines. When combined with the baseline, using Gram-Schmidt enhances the performance mainly in setting (ii) with a linear classifier.

For frameworks (i) and (ii), we use code adapted from (Yehuda et al., 2022; Hacohen et al., 2022), which is based on prior work (Van Gansbeke et al., 2020; Munjal et al., 2020). For the semi-supervised setting (iii), we employ code of the Semi-Supervised benchmark (Wang et al., 2022). Our code is attached to the paper.

We compare our method with other subset selection methods: Uniform at random, which is a competitive baseline, TypiClust (Hacohen et al., 2022) and ProbCover (Yehuda et al., 2022). We do not include other subset selection methods since many of them do not surpass the random selection and others are only slightly better than random selection, as demonstrated in (Guo et al., 2022; Hacohen et al., 2022; Chen et al., 2022).

We tested our method on CIFAR-10, CIFAR-100, Tiny-ImageNet, ImageNet, Yelp and OrganAMNIST (Yang et al., 2023) datasets (see Appendix B for additional details) . Specifically, unless otherwise specified, we utilized SimCLR (Chen et al., 2020) embeddings for CIFAR-10/100, Tiny-ImageNet and OrganAMNIST. For ImageNet, we utilized DINO (Caron et al., 2021) embeddings. We use the SimCLR implementation from (Van Gansbeke et al., 2020) and train ResNet-18 with an MLP projection layer for 500 epochs. Post-training, we extract the 512-dimensional features from the penultimate layer. Our optimization is conducted with SGD with momentum and an initial learning rate of 0.4 with a cosine scheduler. We employ a batch size of 512 and a weight decay of $10^{-4}$. We augment the data with random resize and crop, random horizontal flips, color jittering, and random gray-scale. For DINO, we use ViT-S/16 model pre-trained on ImageNet.

In frameworks (i) and (ii), we evaluated the methods with varying numbers of examples: $[b, 2b, ..., 6b]$, where $b$ represents the number of classes. We report the average results over 3 seeds. We focus on extremely small subsets and we compare our method with baselines that performs well with these sizes (Hacohen et al., 2022; Yehuda et al., 2022). For the semi-supervised framework, (iii), we utilized $b$ labeled examples.

In the fully-supervised framework for CIFAR-10/100, Tiny-ImageNet and OrganAMNIST, we train the model for 200 epochs using the SGD optimizer with Nesterov momentum and a cosine learning rate with an initial step-size of 0.025. We utilize a batch size of $\max\{\#\text{labeled}, 100\}$ and a weight decay of $3 \times 10^{-4}$. Augmentations include random crop and random horizontal flip. For ImageNet, we use the same hyper-parameters as described above, with the exception that we train for 100 epochs and employ a batch size of 50 due to computational constraints. We choose the best epoch according to a validation test. For experimenting with Yelp, we fine-tune a pre-trained BERT (Devlin, 2018). We train for 100 epochs with AdamW optimizer and initial learning rate of $5 \times 10^{-5}$. Instead of selecting the epoch with the highest validation accuracy, we evaluate the test set accuracy based on the last epoch. This choice reduces the computational overhead of repeatedly calculating validation accuracy for a large dataset and model. Additionally, due to limited GPU memory we do not include results with ProbCover. For the unsupervised features we use the embedding of the mid token of each example. This approach avoids the extremely high dimensionality that would result from unfolding the embeddings of all tokens. We employed a single NVIDIA RTX-2080 GPU to undertake the learning processes, encompassing both the acquisition of self-supervised features and training with small subsets of examples.

For the semi-supervised framework with a linear classifier ,(ii), we utilize the features of the labeled data and train a $d \times C$ linear classifier, where $d$ is the dimension of the features and $C$ is the number of classes. To train the classifier, we use a learning rate of 2.5 and 400 epochs. For the semi-supervised framework (iii), we train FlexMatch and SimMatch with a Wide-ResNet-28-10 model using SGD with momentum for 1000k iterations. We employ a learning rate of 0.03, a batch size of 64, and a weight decay of $5 \times 10^{-4}$. Weak

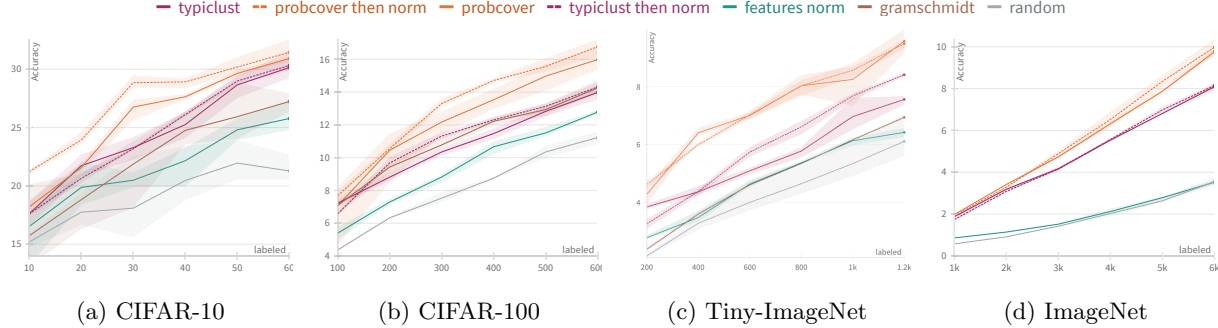

Figure 2: **Fully-supervised framework:** The performance comparison includes both of our methods, randomization with feature norms and Gram-Schmidt, random selection, the baselines TypiClust and ProbCover, and the addition of our norm criterion to these baselines. An average of 3 seeds is presented and the shaded areas correspond to the standard error.

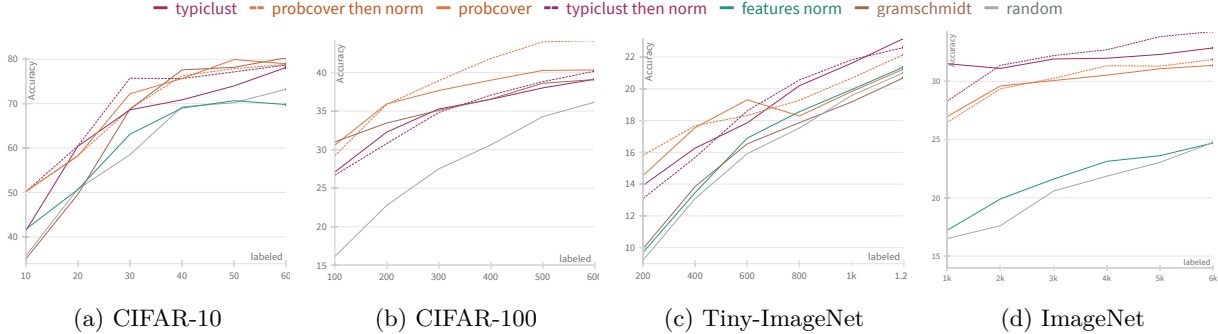

Figure 3: **Semi-supervised with linear classifier:** Results with a linear classifier trained on top of self-supervised features

augmentations such as random crop and random horizontal flip are applied, while strong augmentations are obtained using RandAugment (Cubuk et al., 2020).

### 4.1 Results

**Fully supervised results.** The results in Fig. 2 demonstrate that the simple baseline of randomization according to norm achieves a significant performance boost compared to uniform random selection in CIFAR-10, CIFAR-100 and Tiny-ImageNet. The results also show that our proposed algorithm contributes to CIFAR-10/100. The Gram-Schmidt method contributes an additional enhancement in performance compared to randomization based on norm alone. This is especially notable for CIFAR-100, where the results are comparable with TypiClust. Integrating the norm criterion on top of previous methods yields comparable and even better results, which is particularly evident with CIFAR-10/100 with ProbCover and with Tiny-ImageNet with TypiClust. For ImageNet, using the norm criterion lead to comparable results with the baselines. The results for OrganMNIST are presented in Fig. 5a. In this case, using the norm criterion improved accuracy when used alone and when is combined with ProbCover. Yet, it did not improve results with TypiClust. Note also that the Gram-Schmidt technique usually outperforms random choice but not other baselines tested.

For Yelp (Fig. 5b), the results exhibit variance, likely due to the imbalanced nature of the chosen subset and the unbalanced test set. Selecting an extremely small subset may lead to low label coverage. Additionally, while the training set is balanced, the test set is highly imbalanced, which can amplify the effect of under-represented labels and result in significant fluctuations in test accuracy. We see that on average, the norm and Gram-Schmidt surpass the performance of random choice and that using TypiClust with norm is on-par and usually better then without the norm addition.

Table 2 summarizes the average accuracy results across all datasets in the fully-supervised setting, averaged over all subset sizes. The results demonstrate that incorporating the norm-based criterion consistently enhances performance. In all tested cases, except for TinyImageNet, applying the norm criterion alongside baseline methods yields the highest average accuracy. Moreover, the performance degradation is minimal, with

| Method | CIFAR-10 | CIFAR-100 | TinyImageNet | ImageNet | OrganMNIST | Yelp |
|---|---|---|---|---|---|---|
| Random | 19.12 | 8.09 | 4.26 | 1.88 | 40.40 | 24.26 |
| Norm | 21.60 | 9.41 | 4.81 | 1.99 | 44.05 | 27.50 |
| Gram-Schmidt | 22.61 | 11.10 | 4.52 | - | 41.66 | 29.30 |
| TypiClust | 24.45 | 10.78 | 5.61 | 4.94 | 46.58 | 24.26 |
| TypiClust + norm | 24.48 | 11.22 | 6.02 | 4.93 | 45.79 | **30.85** |
| ProbCover | 25.75 | 12.73 | **7.32** | 5.69 | 45.41 | - |
| ProbCover + norm | **27.43** | **13.10** | 7.29 | **5.76** | **47.99** | - |

Table 2: Average performace of the 6 budgets used for the fully-supervised setting. See the average benefit of using norm.

| | Random | Features Norm | ProbCover | ProbCover + Norm | ProbCover + GS |
|---|---|---|---|---|---|
| FlexMatch | 61.66 | 63.84 | 63.7 | 65.51 | **79.9** |
| SimMatch | 38.03 | 39.68 | 54.68 | 57.08 | **76.02** |
| FlexMatch | 17.81 | 22.8 | 35.68 | 33.93 | **40.04** |
| SimMatch | 17.1 | 22.3 | 36.34 | 34.56 | **40.88** |

Table 3: Semi-supervised training, FlexMatch (Zhang et al., 2021a) and SimMatch (Zheng et al., 2022b), with CIFAR-10 (top) and CIFAR-100 (bottom). Using feature norms is better than uniform random choice and adding GS on top of ProbCover enhances accuracy.

the largest difference being only 0.03. In contrast, the smallest observed improvement is 0.27 for CIFAR-100, with even larger gains achieved for other datasets, highlighting the substantial benefits of the norm-based criterion.

In Appendix H, we present results with larger subsets. In these cases, incorporating the norm criterion leads to comparable or improved accuracy.

Fig. 5a presents results with OrganAMNIST. The findings suggest that incorporating the norm criterion and Gram-Schmidt algorithm is advantageous particularly for higher budgets. Overall, even though we enhance performance with extremely small labeled sets, there is still a significant drop in performance compared to the more than 90% accuracy achieved with the full dataset. Thus, although we improve over the state-of-the-art, there is still room for improvement.

**Semi-supervised learning with a linear classifier.** Fig. 3 displays the results obtained with the semi-supervised framework, where only 1 layer of the neural network is optimized. The results indicate that norm randomization performs similarly to or slightly better than uniform randomization. Our subset selection method shows improvements primarily over norm randomization, especially with higher budgets for CIFAR-10 and with CIFAR-100, where the results are comparable with the TypiClust method. Additionally, integrating the norm criterion with the baselines, ProbCover and TypiClust, generally enhances their performance. It is noteworthy that the results surpass those of the fully supervised setting, indicating that the features used are highly informative. This information can significantly enhance performance, even when only a limited set of labels is available.

**Semi-Supervised with full fine-tuning.** The results for semi-supervised training, where all the network is fine-tuned and pseudo-labeling is employed, are provided in Table 3. Our findings demonstrate a clear improvement in accuracy when utilizing the norm criterion. Randomization according to norm boosts performance and achieves better accuracy than the sophisticated ProbCover method in some cases. Additionally, it appears that incorporating the Gram-Schmidt algorithm on top of ProbCover further enhances performance in this setting. The results do not include TypiClust due to limited availability of computing resources.

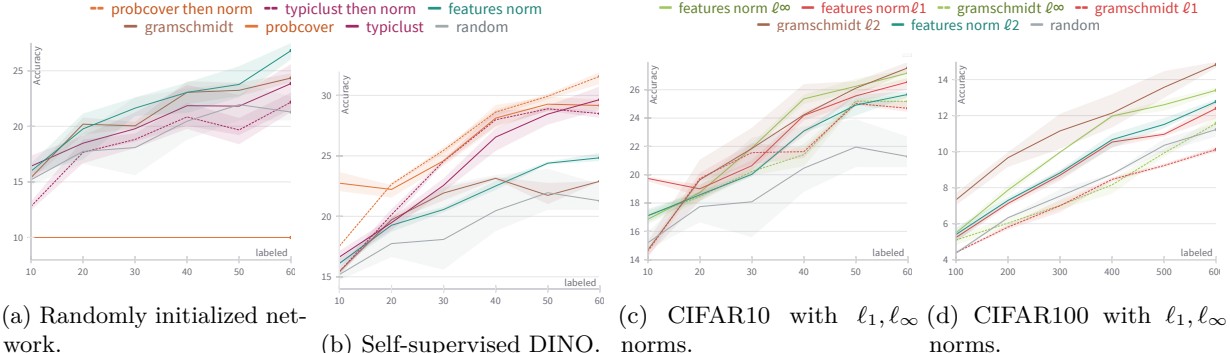

(a) Randomly initialized network.

(b) Self-supervised DINO.

(c) CIFAR10 with $\ell_1, \ell_\infty$ norms.

(d) CIFAR100 with $\ell_1, \ell_\infty$ norms.

Figure 4: **Feature domain.** Figs. 4a and 4b include the results with randomly initialized NN and DINO features. They show that the benefits of the feature norm apply to various feature domains. **Norms type.** Figs. 4c and 4d compares the results of using randomization based on feature norm and Gram-Schmidt algorithm, where the $\ell_1$ and $\ell_\infty$ are used instead of $\ell_2$ for the norm. Gram-Schmidt with $\ell_2$ provides the best results. For feature norm-based selection, $\ell_2$ and $\ell_\infty$ are the best. The experimets are performed in the fully supervised setting.

| Set size | Random | Feature Norms | Gram-Schmidt | TypiClust | TypiClust+Norm | ProbCover | ProbCover + Norm |
|---|---|---|---|---|---|---|---|
| 20 | 271.62 | **251.01** | 267.2 | 244.81 | **240.68** | 253.29 | **250.31** |
| 40 | 215.95 | 214.48 | **213.52** | 211.66 | **209.56** | 207.85 | **207.28** |
| 60 | 188.98 | 188.39 | **186.63** | 185.88 | **184.96** | 182.04 | **179.89** |

Table 4: **FID scores:** Measure distributions similarity between subsets and the remaining training set. Low scores indicate higher similarity. Subsets taken from CIFAR-10.

**Distribution of selected examples.** To include additional evaluation rather than the accuracy, we test the alignment of our selected subset with the training set. To this end, we employ Fréchet Inception Distance (FID) (Heusel et al., 2017) scores, originally presented for numerical evaluation of generative models. FID measures the similarity between the distributions of artificially generated images and real images. We use it to measure the similarity between the chosen subsets and the remaining examples in the training set. Lower FID scores, as detailed in Table 4, indicate a closer match to the original dataset, are particularly evident when utilizing the norm criterion. We calculate the FID scores with Seitzer (2020). Our approach results in a distribution that closer to the original dataset. In Appendix L, we provide qualitative examples chosen with the norm criterion to show that images with high norms are more likely to be easier to recognize for humans.

## 4.2 Empirical Validation of Assumptions

**Correlation of accuracy and norm.** Instead of forcing high norm explicity, we test the correlation between performance and features norm in randomly selected sets. Fig. 7 shows the accuracy after training on CIFAR-10 subsets in a fully-supervised framework. To create the figure, we randomly sampled 100 sets of 200 and 300 labeled images and 500 sets of 20 and 30 labeled images and trained a network for them in a supervised manner. For each subset, we present the accuracy as a function of its average feature norm. Having the scatter plot, we added a regression line over the points, which demonstrates that high accuracy correlates with high feature norms, which justifies the assumption we make in this work. In Appendix J, we provide results obtained with DINO features to show that the phenomena is not related to a specific unsupervised model. Although in some cases the correlation is weaker, there is still a consistent positive correlation between high accuracy and high norm across all tested cases. The low correlation between feature norm and accuracy suggests that while feature norm is useful, as our experiments show, it can greatly benefit from complementing it with other methods that depend on features. In our work, we proposed using Gram-Schmidt orthogonalization to reduce correlations among features within the chosen subset and also combined our approach with other existing methods. Note also that as we have shown the norm criterion can be embedded in existing subset selection methods and improve them.

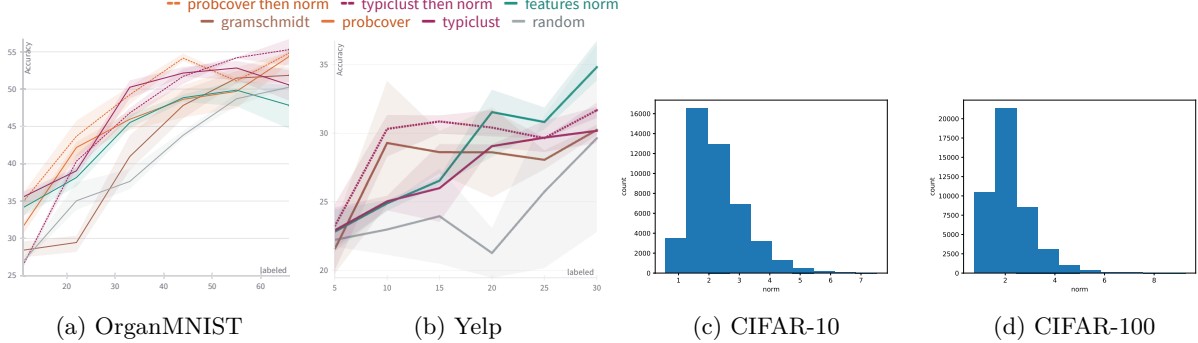

(a) OrganMNIST     (b) Yelp     (c) CIFAR-10     (d) CIFAR-100

Figure 5: Figs. 5a and 5b include results in the fully-supervised setting with OrganMNIST and fine-tuning of BERT with Yelp. Notice that also in this case, our approach provides benefits for subset selection. Figs. 5c and 5d present the histograms of the feature norm with self-supervised features. The norms are diverse, indicating that the distribution they induce is not vacuous.

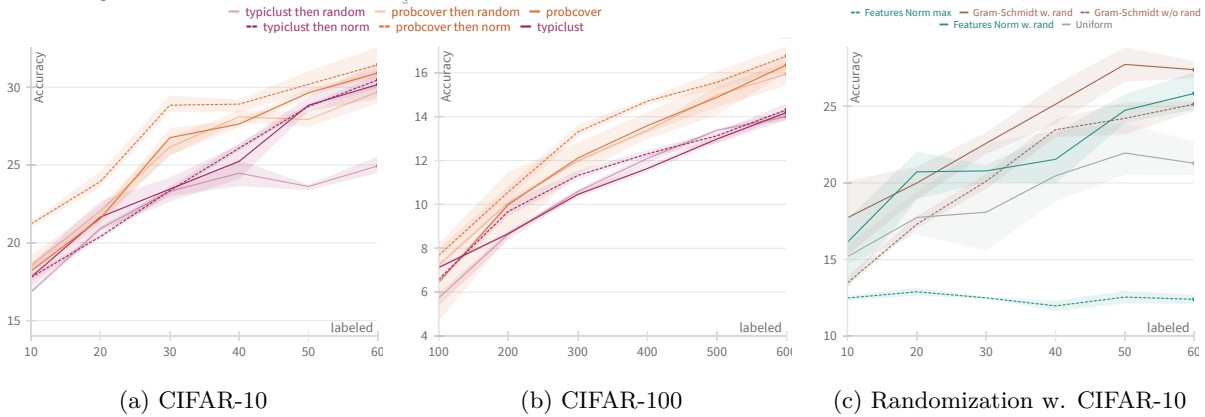

(a) CIFAR-10     (b) CIFAR-100     (c) Randomization w. CIFAR-10

Figure 6: Figs. 6a and 6b include results obtained with uniform randomization rather than randomization according to the feature norm. They show that randomization based on feature norms is pivotal for the enhancement in performance. Fig. 6c compares selecting examples according to the feature norm or Gram Schmidt with and without randomization. Note that using randomization is beneficial. The experimets are performed in the fully supervised setting.

**Distribution of feature norm.** In order to demonstrate the sampling distribution, we present in Fig. 13 the histogram of feature norms across the entire datasets. Specifically, we illustrate the feature norms of the SimCLR features for the CIFAR-10/100 datasets. The plot reveals diverse values of feature norms, indicating a non-vacuous distribution from which examples are sampled.

### 4.3 Ablation Study

**Dependency on the feature embedding.** To ensure that the benefits of our method are not dependent on a specific feature domain, we conducted experiments with other embeddings. Specifically, we employed a self-supervised approach, DINO (Caron et al., 2021), which is known for its informative features. We experimented with the fully-supervised framework with CIFAR-10. The results in Fig. 4b indicate that utilizing norm randomization is beneficial. Moreover, the addition of norm on top of existing methods mostly enhances the performance. Also, for the Gram-Schmidt method, we observe a performance gain, particularly at low budgets. In Appendix C, we include results on CIFAR-100.

Given the computational demands and potential unavailability of pre-trained self-supervised models, we conducted experiments with features induced by randomly initialized neural networks with CIFAR-10. The results with initialized neural networks are presented in Fig. 4a. We observed that the performance of ProbCover collapsed and suffered from accuracy comparable to a random classifier.

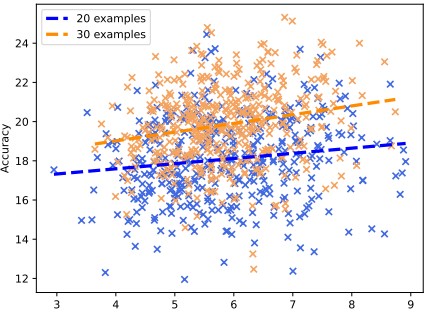 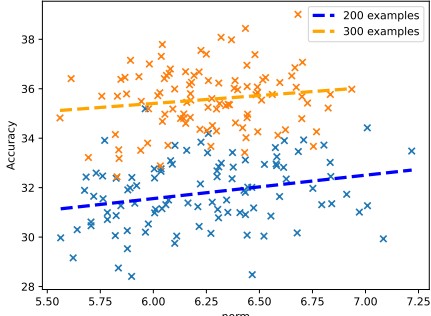

Figure 7: **Correlation with Norm:** Test Accuracy of models trained with 20/30 examples (left) and 200/300 examples (right) uniformly sampled from CIFAR-10 as a function of their self-supervised feature norms. The regression lines indicate a positive correlation between norm and accuracy. Although the correlation is not high, it remains consistent across subset sizes and self-supervised features (see results with DINO in Fig. 16).

The benefits of TypiClust over random sampling are limited in this case, and using the norm criterion did not improve this. It is noteworthy, that this is consistent with the claim made by Yehuda et al. (2022), which clearly stated that TypiClust and ProbCover rely on informative features and suffer from poor performance with the RGB space. Our approach demonstrates benefits in both scenarios. Employing the Gram-Schmidt algorithm yields benefits over random sampling and, notably, using the norm criterion alone induced high accuracy. Even though other methods experience a degradation in accuracy when using less informative features, we observe that randomization according to norm maintains its performance. Appendix C includes results with CIFAR-100 and the raw RGB space.

**Othen norm types.** Given that some pruning methods are related to norms other than the $\ell_2$ norm (Li et al., 2017), we present results with $\ell_1$ and $\ell_\infty$ norms with CIFAR-10 and CIFAR-100 in Figs. 4c and 4d. We observe that randomization according to $\ell_\infty$ yields good results. The performance of Gram-Schmidt with norms other than the $\ell_2$ norm harms performance, suggesting that our method suits the Euclidean norm. Randomizing according to $\ell_1$ is better than $\ell_2$ for CIFAR-10 but slightly worse for CIFAR-100. We employ the $\ell_2$ norm due to its consistent performance across scenarios. In Appendix D, we include results with ProbCover and TypiClust.

Additionally, we tested the effect of normalizing the feature norms by the norm of the input. The normalization did not significantly change the results, so we have not included them in this paper.

**Necessity of randomization.** We investigated the benefits of randomization for successful subset selection. In Fig. 6c, we present results, comparing scenarios where examples with the maximal norm are chosen based on feature norms and with our algorithm. Specifically, in our algorithm (Algorithm 1), the randomization step is replaced with selecting the example with the highest feature norm. The results indicate that for selection based solely on norms, randomization is crucial for achieving accuracy gains, as accuracy only marginally surpasses that of a random classifier.

**Model Agnostic Selection.** To demonstrate the model-agnostic nature of our method, we include results from fine-tuning the selected subsets using MobileNet (Howard, 2017). These results further confirm that the norm criterion consistently enhances performance, even in this scenario. The results appear in Fig. 12.

**Replace the norm criterion with random sampling.** To better assess the gain in performance for Typiclust and ProbCover that is achieved by the addition of our norm criterion, we replace the norm criterion with random sampling. Figs. 6a and 6b contain the results with CIFAR-10/100. The results indicate that the use of random sampling with the sets chosen by TypiClust and ProbCover leads to either degradation or comparable results. This is in contrast to the benefits of random selection based on the features' norm values. A list of references to results figures and their settings is included in Appendix A.

### 4.4 Discussion and Limitations

The results presented above highlight that employing a simple criterion such as the norm can yield non-trivial performance. Importantly, the norm criterion is easy to integrate into existing methods, usually enhancing or maintaining competitive performance with minimal additional computational overhead. Yet, in some minority cases, our approach leads to a small degradation in performance.

The Gram-Schmidt approach also shows improvement compared to uniformly random subset selection and norm-based selection. However, it falls short of the performance achieved by more sophisticated baselines, albeit with the advantage of lower computational complexity.

The empirical motivation for using norm values was tested (Appendix J), revealing that while the norm shows a low correlation with accuracy, its correlation remains stable across different feature domains and subset sizes.

Through an extensive ablation study, we validated the robustness of our approach to variations in the norm order and feature domains. Our analysis confirms that the norm is indeed a beneficial measure for randomly chosen subsets and that the distribution of the norms carries meaningful information, rather than being vacuous.

## 5 Conclusion and Future Work

In this work, we have presented a novel approach to addressing the subset selection problem for deep neural network training. Leveraging insights from neural network pruning and focusing on the norm criterion of network features, we have introduced a method that significantly enhances the efficiency and effectiveness of subset selection. Our findings show a correlation between the output features' norm and accuracy, highlighting the importance of feature norms in the selection process. Moreover, we have presented a tailored algorithm that combines the norm criterion with the Gram-Schmidt process to ensure coverage of the feature space.

Furthermore, our evaluations across diverse settings and datasets validate the efficacy of our approach. By comparing our method with existing subset selection techniques, such as uniform random sampling, TypiClust, and ProbCover, we have shown consistent improvements in model performance, particularly when dealing with extremely small subsets selected from large unlabeled pools. Our method achieves new state-of-the-art results in many cases, underscoring its relevance and practical utility in scenarios where labeled data availability is limited, such as medical applications. We discuss the broader impact of our suggested approach in the appendices.

Looking ahead, our work opens several avenues for future research. One promising direction is using the connection we draw between pruning and subset selection. In this work we used the norm criterion, but future work could explore the migration of other pruning methods to the subset selection. Additionally, the training data attribution problem is closely related to the subset selection problem (Grosse et al. (2023); Schioppa et al. (2022); Chhabra et al. (2024); Ilyas et al. (2022); Park et al. (2023), partial list). In training data attribution, the goal is to quantify how specific training examples influence the model's predictions at test time and the analysis is typically conducted *after* the training. In contrast, subset selection seeks to identify a representative subset of the data *before* training, often without access to the labels. Despite these differences, insights from attribution methods could be used for subset selection by highlighting examples likely to have a high impact on model performance.

Overall, we believe that we advance the field of subset selection and lays a foundation for developing more effective and informative annotation strategies in deep learning.

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

**Broader Impact Statement**

Our proposed method for effective subset selection has a positive social impact, particularly in resource-constrained environments. By optimizing the selection of training data subsets, this approach reduces the computational and financial costs associated with data annotation, making advanced machine learning models more accessible and efficient. This can democratize access to powerful AI tools in underfunded sectors such as healthcare and education, where data labeling is often prohibitively expensive. Ultimately, the methodology enhances the scalability and applicability of AI, fostering broader societal benefits through more equitable distribution of technology and resources.

# A  References List of the Results

Below we list the settings and references of the results presented in the paper.

| Setting | Features Source | Notes | Location |
|---|---|---|---|
| Fully-supervised | SimCLR | | Figs. 2, 5a and 5b |
| Fully-supervised | SimCLR | Average accuracy | Table 2 |
| Linear | SimCLR | | Fig. 3 |
| Semi-supervised | SimCLR | | Table 3 |
| Fully-supervised | DINO | | Figs. 4b and 8b |
| Fully-supervised | Random NN | | Figs. 4a and 8a |
| Fully-supervised | RGB | | Fig. 9 |
| Fully-supervised | SimCLR | Ablation: Norm order | Figs. 4c, 4d and 10 |
| Fully-supervised | SimCLR | Ablation: Randomization | Fig. 6 |
| Fully-supervised | SimCLR | Ablation: Model Agnostic | Fig. 12 |
| Fully-supervised | SimCLR | Baseline + Gram-Schmidt | Fig. 11 |
| Fully-supervised | SimCLR | Initial pool | Fig. 13 |
| Fully-supervised | SimCLR | Large Subsets | Fig. 14 |

# B  Additional Datasets Details

**OrgansMNIST**. In order to assess the benefits of our approach, which involves acquiring specialized and potentially costly labels, we utilize OrganAMNIST Yang et al. (2023), comprising abdominal CT images. It consists of 34,561 training images and 17,778 test images, categorized into 11 classes.

**Yelp.** We include results on the Yelp dataset, which consists of user reviews and corresponding ratings. This classification dataset features five possible ratings and includes a balanced training set of 650,000 examples. Additionally, the test set contains 50,000 reviews.

# C  Results with other features domains

Additional results incorporating various feature types with CIFAR-100 are presented in Fig. 10. Notably, when utilizing features induced by an initialized neural network, randomization according to norm yields a substantial performance boost compared to uniform random selection. For ProbCover, the results surpass those of random selection, while for TypiClust, the performance with and without norm criteria remains comparable, with limited gains over random selection. Gram-Schmidt, however, exhibits enhanced performance across the board.

In the case of DINO features, results from Gram-Schmidt, randomization according to norm, and uniform random selection are comparable. Using the norm criterion on top of the baselines leads to comparable results.

We include results with raw RGB values of the image used as features. The results are presented in Fig. 9 and demonstrate advantages for both norm randomization and Gram-Schmidt algorithm compared with the

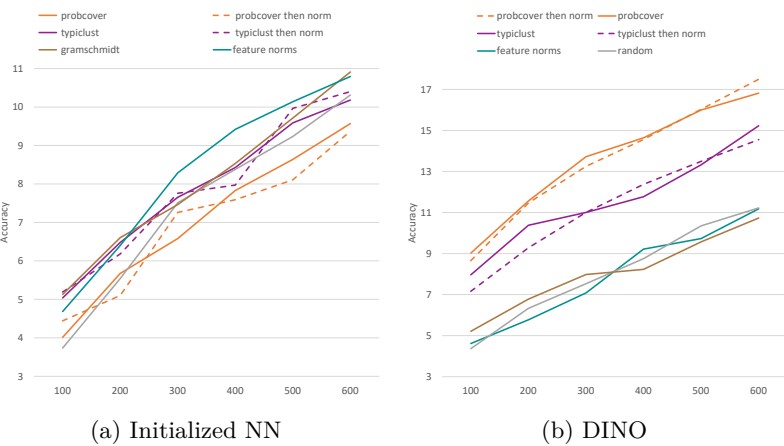

(a) Initialized NN        (b) DINO

Figure 8: Accuracy results with CIFAR-100. **Right:** Features of randomly initilialed NN. **Left:** Features with self-supervision model of DINO.

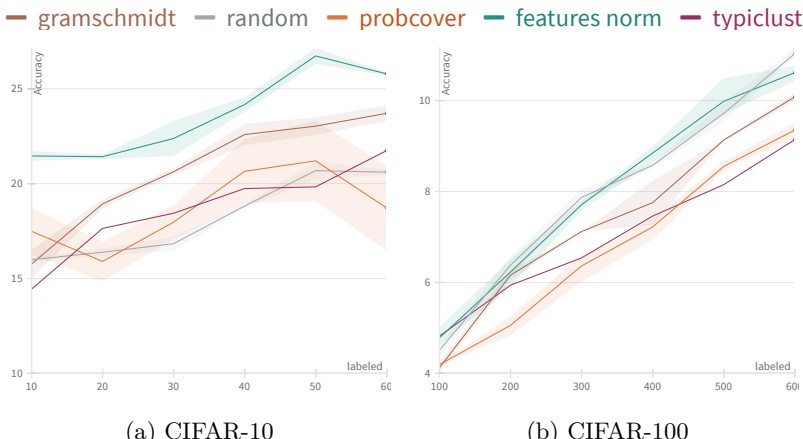

(a) CIFAR-10        (b) CIFAR-100

Figure 9: Results with raw RGB values.

baselines with examples chosen from CIFAR-10. For CIFAR-100, norm randomization in comparable with uniform randomization and our method do not surpass the effective random baseline. Additional baselines also exhibit performance lower than the random selection.

## D   Results with additional norm types

To evaluate the impact of different norm types, we conducted experiments on CIFAR-10 and CIFAR-100 datasets using TypiClust and ProbCover as baselines. We incorporate $\ell_\infty$ and $\ell_1$ norms, the results are presented in Fig. 10. With CIFAR-10, employing $\ell_1$ norms did not consistently demonstrate clear benefits and, in some cases, even degraded the results when added to the baselines. Conversely, the results with CIFAR-100 were comparable to the baselines. Regarding $\ell_\infty$ norms, integrating them improved results with ProbCover for CIFAR-10 but had adverse effects with TypiClust. However, with CIFAR-100, the results were comparable with ProbCover and showed enhanced performance with TypiClust. Considering $\ell_2$ norms, there was a notable increase in accuracy with ProbCover and comparable results with TypiClust for both CIFAR-10 and CIFAR-100. Overall, utilizing $\ell_2$ norms exhibited the most stable performance across both datasets, leading us to adopt it consistently throughout our work.

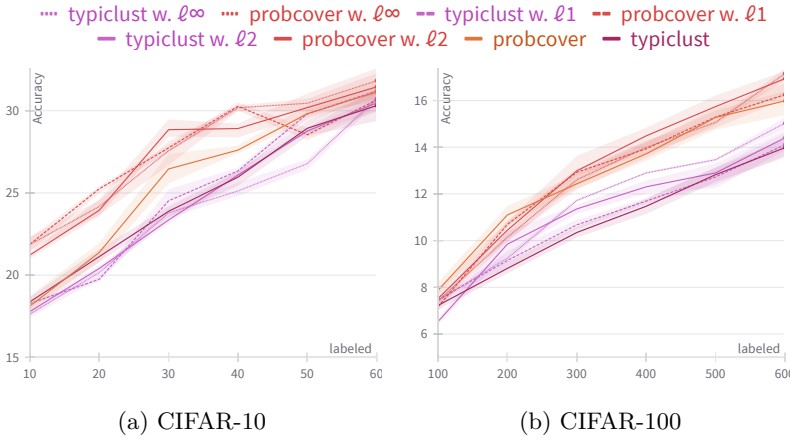

Figure 10: Accuracy results with multiple types of norms added on top of baselines.

# E    Results with Gram-Schmidt on top of baselines

In Fig. 11, we present results incorporating ProbCover and TypiClust with the Gram-Schmidt method. We select examples by choosing twice the required budget and then utilizing our suggested algorithm (see Algorithm 1). We observe that our method yields comparable results with ProbCover, but slightly degrades the results with TypiClust in the fully-supervised framework. Using Gram-Schmidt in the semi-supervised with linear classifier mostly enhence the performance.

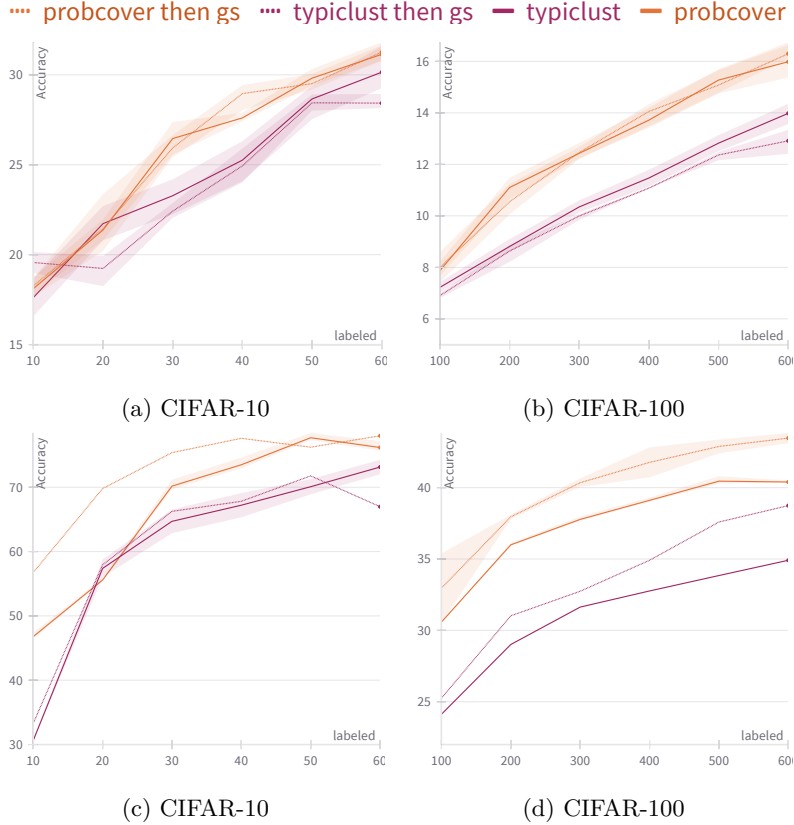

Figure 11: Accuracy results with Gram-Schmidt algorithm (Algorithm 1) added on top of baselines. Results with fully-supervised (top) and semi-supervised with linear classifier (bottom).

# F    Results with MobileNet

In order to asses that the benefits of our methods are model agnostic we include results with fully-supervised setting with MobileNet (Howard, 2017). The results present that our model is using norm criterion indeed boosts performance. For Gram-Schmidt the performance is better or on-par with random subsets.

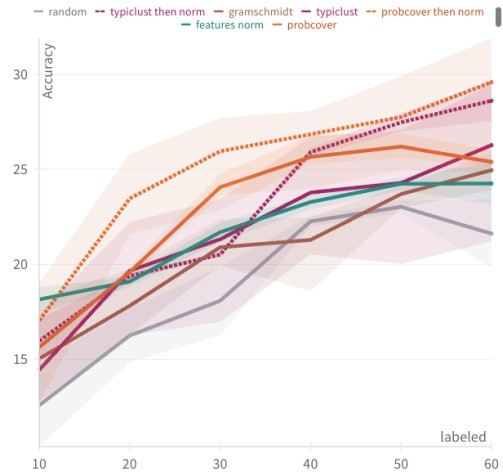

Figure 12: Results with training fully-supervised MobileNet with examples taken from CIFAR-10.

# G    Results with Initial Pool

Previous active learning methods for subset selection typically required an initial non-empty labeled pool for training an initial learner, from which the selection process is bootstrapped. Therefore, we include results in Fig. 13 where training is conducted with an initial uniform randomly selected pool of $b$ labeled samples, where $b$ is the number of classes. The results demonstrate that randomization according to norm remains beneficial for performance even in this scenario, when compared with completely random sampling. While our Gram Schmidt method does not enhance performance beyond randomization according to the norm, it does demonstrate advantages over uniform random selection. The results with the addition of the norm criterion to TypiClust and ProbCover show comparable performance to the baselines. For TypiClust, the norm addition yields improved results. Overall, we observe that also in this case, using the feature norm criterion is beneficial for performance.

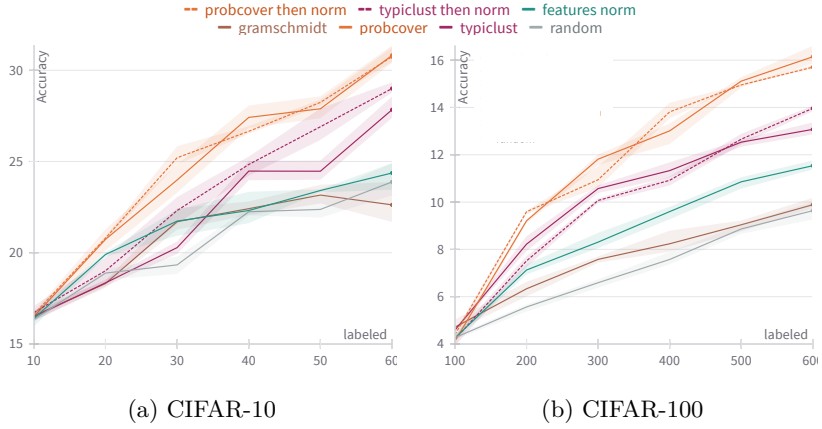

(a) CIFAR-10                                      (b) CIFAR-100

Figure 13: Figs. 13a and 13b include results with an initial random labeled pool. The size of the initial pool is the number of classes. Notice that also in this case, our approach provides benefits for subset selection.

## H   Results with Large Subsets

We tested our approach with larger subsets by selecting 1000, 2000, and 3000 examples from CIFAR-10. The results, included in Fig. 14, demonstrate that using norm randomization surpasses the performance of uniform random selection and Probcover. Additionally, our Gram-Schmidt algorithm yields improvements over uniform random selection. Furthermore, adding norm randomization on top of the baselines leads to comparable results.

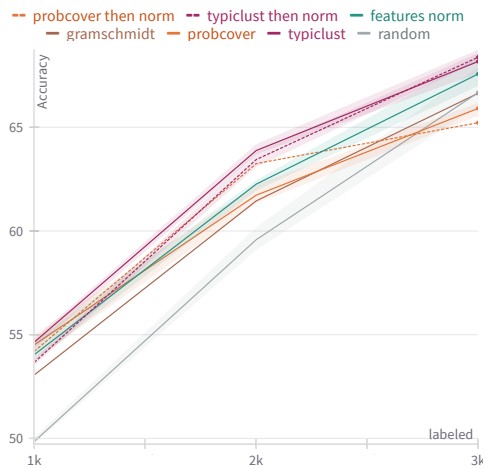

Figure 14: Results with large subsets, of 1000, 2000 and 3000 examples, selected from CIFAR-10.

## I   Network Pruning Norms

To demonstrate the relationship between the norms of the weights and successful pruning, in Fig. 15, we randomly sample 1000 unstructured subnetworks and compute their weight norms. We then compare these norms to those obtained using the well-known iterative magnitude pruning (IMP) method Frankle & Carbin (2018), where weights with low magnitude are iteratively zeroed after training. This method outperforms by a margin the randomly sampled networks and as can be seen in the figure, the IMP weights are much larger than those of random networks accross all the layers.

## J   Correlation of Feature Norms and Accuracy

In Fig. 16, we present additional results showcasing the correlation between accuracy and feature norms. These findings are consistent across small sets (with 20 and 30 examples) as well as larger sets (with 200 and 300 examples). We demonstrate that this correlation persists when using DINO features, providing further evidence of the relationship between feature norms and performance.

## K   Qualitative Results EigenFaces

We include results of 10 images chosen with different subset selection methods. The images are presented in Fig. 17. For random sampling, we see that the images are diverse but there is an uninformative image of a man looking down so his eyes are not shown in the image. For examples with maximal norm, we see the lack of diversity in the images as 2 people have 3 images. The images chosen with randomization according to norm and Gram-schmidt are more diverse and informative than other methods.

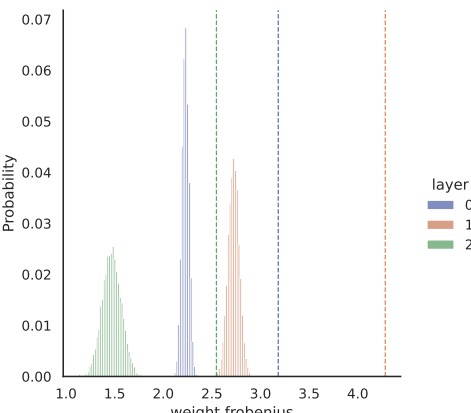

Figure 15: **Pruning bias toward high norm.** The histogram illustrates the Frobenius norms of each layer in 1000 uniformly random sub-networks. The dotted lines represent the norm of the pruned model with IMP Frankle & Carbin (2018), which has higher norms than random and achieves better performance.

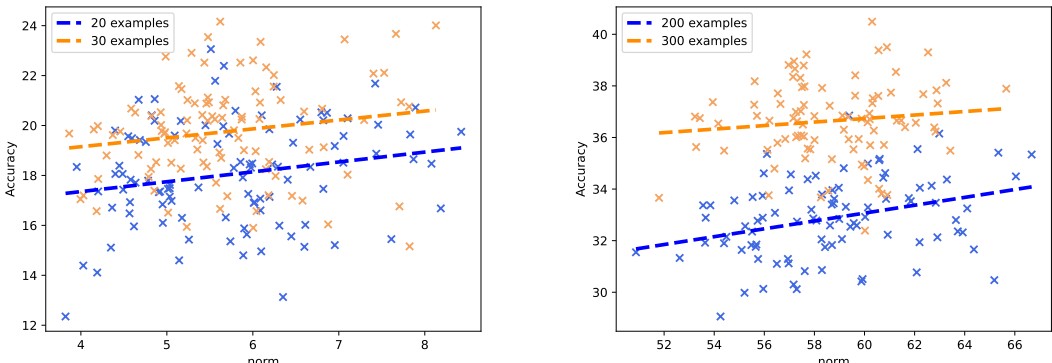

Figure 16: Accuracy of a network trained using 100 uniformly sampled random sets of 20 and 30 examples (left) and 200 and 300 examples (right) from CIFAR-10 as a function of their self-supervised feature norms. Here we present results with DINO. The lines represent the regression line, indicating a positive correlation between norm and accuracy.

## L   Qualitative Results CIFAR-10

To provide a qualitative evaluation, we present randomly chosen images with varying correlations to the norm. These images are divided into three categories: positive correlation (Fig. 20), uniform sampling (Fig. 19), and negative correlation (Fig. 18).

For images with negative correlation to the norm, we use $p_i \propto \max_{j=1,..,N}\{\|F_j\|\} - \|F_i\|$. The low norm images include those that are not typical and are easy to recognize. For example, note the dog in the lower row of the first column, the cat in the second-from-bottom row of the first column, and the bird in the fifth row of the third column. The uniformly sampled images also include some uninformative examples. For instance, see the ship in the first row of the sixth column. However, these images are generally more informative than those with low norms. Finally, for the high norm images, it is difficult to find uninformative examples. These images are generally more easy to recognize and informative.

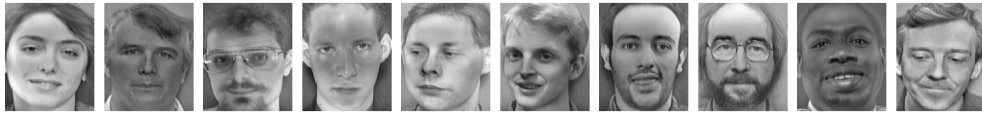

(a) Random

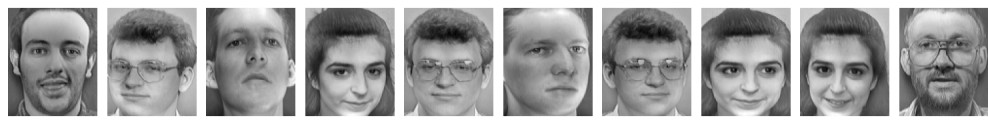

(b) Maximal Norm

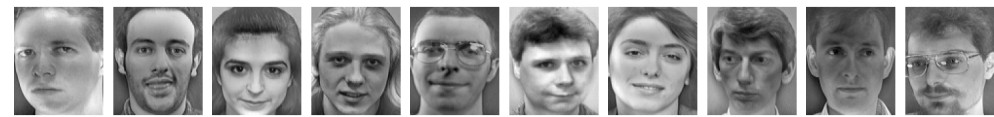

(c) Norm randomizatiom

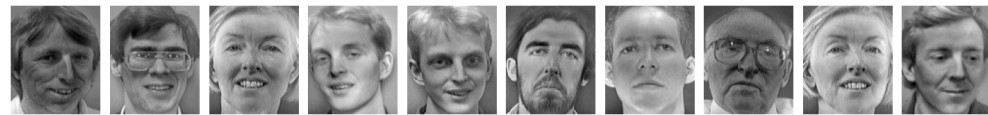

(d) Gram-schmidt

Figure 17: 10 images taken from a subset of examples of AT&T chosen with different methods.

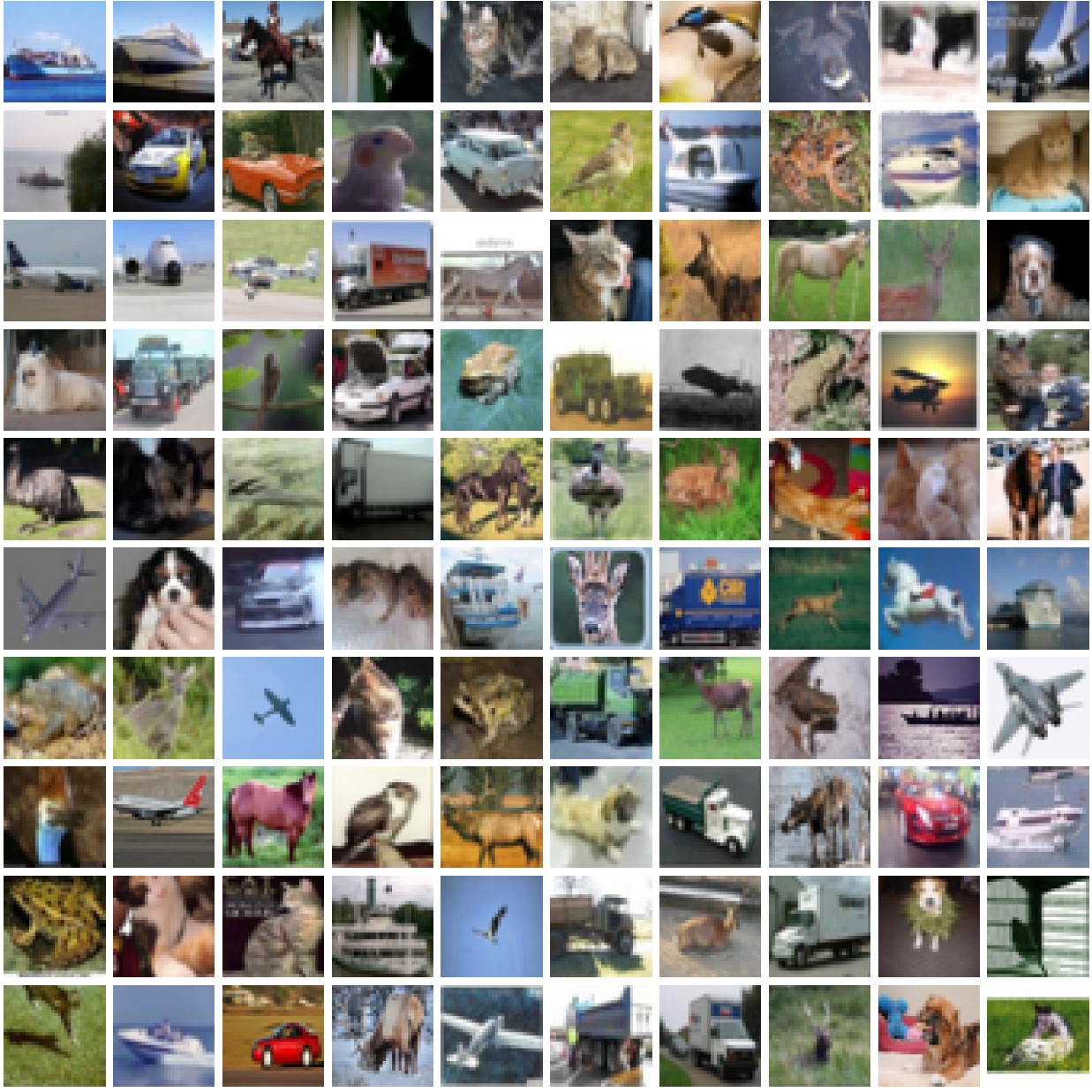

Figure 18: Images chosen randomly with negative correlation to feature norms.

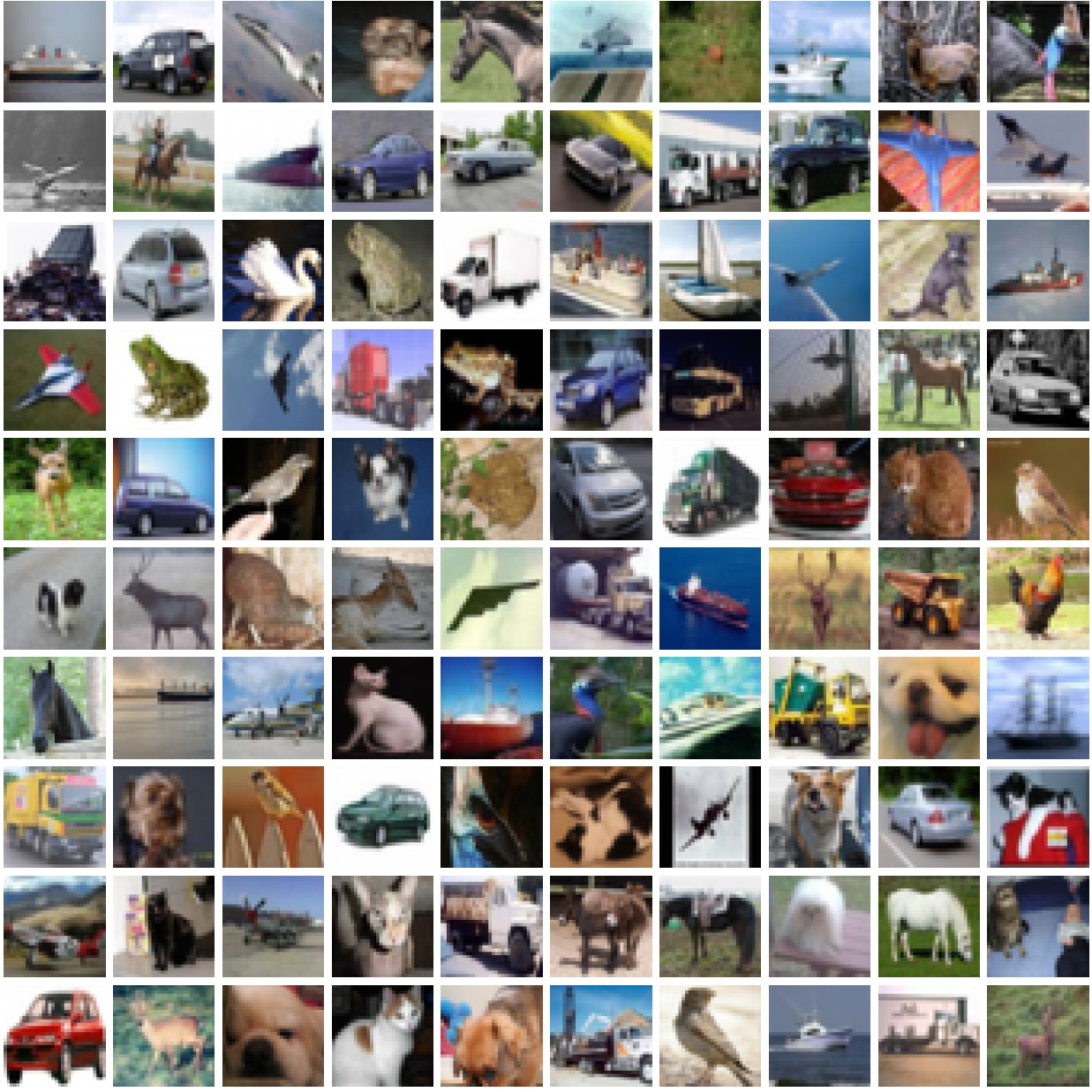

Figure 19: Images chosen uniformly at random.

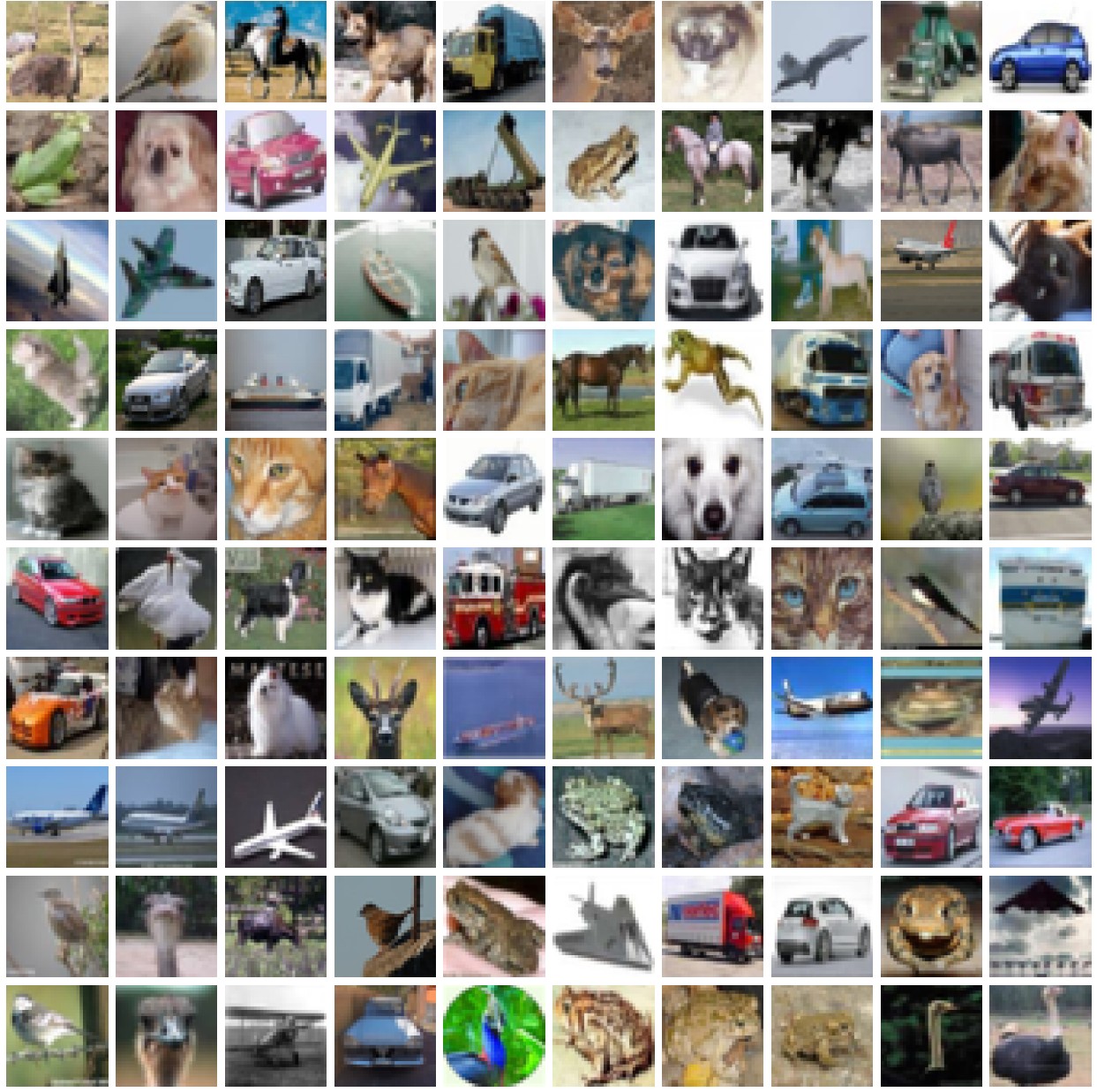

Figure 20: Images chosen randomly with positive correlation to feature norms.

