# OpenReview forum: "Effective Unsupervised Subset Selection Through The Lens of Neural Network Pruning"
_TMLR — Rejected by TMLR_

### Review · Reviewer_WiR4 · 2024-11-16

**Summary Of Contributions:**

This paper addresses the subset selection problem by exploring its relationship with neural network pruning. The authors propose leveraging the norm of feature representations to improve subset selection strategies and introduce a novel combination of norm-based sampling and the Gram-Schmidt process to enhance diversity and coverage. Experiment results evaluated on Cifar-10, Cifar-100, ImageNet, and Tiny-ImageNet demonstrate the effectiveness of the method.

**Audience:**

Yes

**Claims And Evidence:**

No

**Requested Changes:**

- Add a problem statement section.
- Add justifications for the proposed method.
- Add experiments on larger-scale datasets and model backbones.
- Fix typos and improve presentations.

**Strengths And Weaknesses:**

Strength: The proposed norm-based method is simple but effective (at least in the datasets the author experimented on).

Weakness:
- The problem formulation lacks clarity. For example, when annotating only a small subset of the dataset with added randomness, it is unclear how the authors evaluate different subset selection methods. Are fixed validation/test sets predefined for all methods? Additionally, the choice of subset size (s) is not well-explained. Different methods may have varying sensitivity to s, and using a fixed value could unfairly favor certain approaches over others.
- The writing and presentation are also confusing. For example, in Figure 1(b), it appears that norms are computed after the data passes through a model, yet Equation 1 and Section 3 define features as raw data input. Clarification is needed here. In addition, There is a repetitive sentence in the second-to-last paragraph of page 2.
- The proposed method lacks sufficient theoretical justification or empirical validation to establish why norm-based selection is a valid or optimal choice for subset selection. This undermines the method's scientific basis.
- The inclusion of randomization (at the end of Section 3) for performance enhancement diminishes the perceived effectiveness of the norm-based approach. It suggests that the proposed method alone may not be robust enough to consistently perform well.
- Furthermore, while the authors claim the effectiveness of their methods in terms of accuracy, the datasets used are too simple (Cifar-10, Cifar-100, ImageNet, Tiny-ImageNet). Since the proposed method is simple and scalable, its validity should be tested on larger and more diverse real-world datasets with complex model architectures to better establish its generalizability.
- The problem of interest is very similar to the training data attribution problem, which is being explored extensively [e.g. 1-5]. The authors are encouraged to include discussions about that.

1. https://arxiv.org/pdf/2308.03296
2. https://arxiv.org/pdf/2112.03052
3. https://arxiv.org/pdf/2405.03869
4. https://arxiv.org/pdf/2202.00622
5. https://arxiv.org/abs/2303.14186

---

> ### Author Response · Authors · 2024-12-12
>
> We sincerely thank the reviewer for their valuable feedback and constructive suggestions. We have carefully addressed each point and incorporated substantial changes to improve the clarity, presentation, and scope of our work. The revisions are highlighted in blue in the revised manuscript. Below, we provide a point-by-point response to your comments:
>
> 1. Validation and test sets are predefined and shared across all methods for consistent evaluation. We clarify this in the experiment section of the revised manuscript. We chose small budgets for our experiments, as our baselines are known to perform well in this regime. Additionally, we validated that our method also enhances performance with larger subsets, and these results are now included in the appendix.
>
> 2. We revised the manuscript to clarify the relationship between raw data input, extracted features, and norm computation. Specific improvements were made to the caption of Figure 1(b) and the explanation in Section 3 to address this inconsistency. Additionally, a dedicated problem statement section has been added to provide a clear and concise formulation of the problem we address.
>
> 3. We added an additional motivation section, including a simple example demonstrating the validity of norm-based selection using EigenFaces. This example illustrates the relationship between the norm Gram-Schmidt and subset informativeness, providing some theoretical grounding for our approach.
>
> 4. We clarified in Section 3 that randomization serves to enhance diversity within the selected subsets rather than compensate for weaknesses in the norm criterion itself.
>
> 5. We expanded our experiments to include fine-tuning with BERT on the Yelp dataset (650,000 training examples) and training with MobileNet. These results are included in the experiment section and appendix, demonstrating the scalability and robustness of our method.
>
> 6. We present the average performance across multiple subset sizes for all tested datasets in Table 2. The results clearly demonstrate that, on average, incorporating the norm criterion is highly beneficial.
>
> 7. We added a discussion of the training data attribution problem in the future work section, including relevant references (e.g., [1-5]) to better situate our work within the existing literature.
>
> We hope these changes address the reviewer’s concerns.

---

> > ### Comment · Reviewer_WiR4 · 2024-12-21
> >
> > I appreciate the authors' efforts in addressing some of my concerns. I have one additional clarification question:
> > - In the revised manuscript, Section 3 mentions that $F_i$ is "usually derived from an unsupervised trained model." Could you clarify what this "unsupervised trained model" refers to? Specifically, do you first train a model to learn the distribution of the data, or do you use a randomly initialized model? Additionally, is this the same model used during the training phase, or is it distinct? Can you provide details on this particular choice of models? In algorithm 1, this model should be noted as input.

---

> > > ### Author Response · Authors · 2024-12-21
> > >
> > > Thank you for your interest. We are delighted to provide further details of our settings.
> > >
> > > The model used for feature extraction is typically different from the one ultimately trained in a supervised manner after selection. Below, we provide a detailed explanation of the settings.
> > >
> > > 1. We derive features from unsupervised models, specifically SimCLR and DINO. For DINO, we utilize the unsupervised pre-trained model based on ViT-16, trained with the Google Landmark v2 dataset. For SimCLR, we train an unsupervised model for each dataset using ResNet-18, adhering to the implementation details of Typiclust and ProbCover. After deriving features, we select a small subset to label. Finally, during the training step, we train a randomly initialized ResNet-18 from scratch using only the labeled data.
> > >
> > > 2. In the ablation study, we additionally use a ResNet model initialized without training to extract features, which is subsequently fine-tuned with the subset of labeled data.
> > >
> > > 3. For the Yelp dataset, we employ a pre-trained BERT model to extract features, the pre-training did not involve Yelp data. We then fine-tune the same BERT model on the Yelp dataset.
> > >
> > > 4. Finally, the model used to derive the features and the feature extraction step are incorporated to Algorithm 1.
> > >
> > > We hope this provides a comprehensive response to your question and enhances your understanding of our methodology.

---

### Review · Reviewer_zjdK · 2024-11-19

**Summary Of Contributions:**

This work applies principles from neural network pruning methods for the purpose of data sample selection for annotation. The selection of data for annotation is an important problem for a number of domains in which annotation is expensive and data is limited in scale. This work proposes the perspective of data samples as if they were sub-components of a neural network and, as such, could be given an importance score and pruned (or retained) based upon this score. Borrowed from network pruning, this work proposes the measurement of the L2 norm of feature (data) samples and a re-weighting of the selection of these samples based upon this norm. Furthermore, they propose the addition of an orthogonalizing process to reduce the weighting of samples based upon their similarity to existing selected samples. This allows the selection of an entire subset of data samples, for the purpose of annotation and training. The impact of the addition of such components to the sample-selection pipeline during training neural networks is explored in the results section.

**Audience:**

Yes

**Claims And Evidence:**

No

**Requested Changes:**

The changes here are all required for an acceptance recommendation.

## Typos and errors
- Citations throughout: In many cases, the citations are incorrectly formatted and should be ideally cited in some form of brackets. Please check the use of commands such as `\cite{...}` vs `\citep{...}`, `\citet{...}` and more.
- Section 2, Paragraph 3: a sentence is repeated twice back to back.
- Typos: Please check the work once again for typos. A few found: 'discusse' vs 'discuss', 'aproach' vs 'approach' and 'appendix' vs 'appendices' in Section 5

## Major changes
- **Reformatting of results**: as described in the weaknesses section, the results should be reformatted for a better narrative flow and a discussion section added. The implications of each set of results and how these results should be taken forward in the field should become more clear.
- **Results clarity**: the true impact of this current method needs to be more clearly and honestly represented - especially its failure to provide above baseline performance in a range of cases. To truly demonstrate impact, more seeds would be required to better identify trends.
- **Methods Clarity**: the current method section does not express well how the novel propositions (norm and gramschmidt) are combined with existing methods (TypiClust and ProbCover). Instead a cursory statement is made about their conclusion. It should also be made clear whether the novel propositions can be combined arbitrarily with alternative proposals for subset selection.

Please see the Weaknesses section above for any further clarity on why these major changes have been requested.

**Strengths And Weaknesses:**

## Strengths:
- **Strong Introduction**: This work begins with a strong first half, well describing the state of the field and the construction of scores and weights for each sample in a dataset
- **Extensive Experimental Regimes**: The experiments section tests this method across a range of datasets, as well as network/task constructions, with multiple repeats for most plots as well as sub-experiments which analyse particular aspects of the theoretical proposal (e.g. Figure 7)

## Weaknesses:
- **Weak empirical results**: Though the experiments section covers extensive ground, the value of the proposed method is difficult to determine. In particular, for Figures 2, 3, and 4, the proposed methods are often on-par or worse than their counterparts (e.g. `typiclust` vs `typiclust + norm` but also all other combinations). The provided error bars (where present) often also show significant overlap, indicating the the benefit of the proposed methods is not necessarily significant. This in general reduces the value and potential impact of the work as it is currently presented.
- **Results section structure**: The results section does contain a wide range of results, however these are structured such that they are often multiple pages away from where they are referenced in the text. Furthermore, the flow of results does not have a strong logical narrative structure. This could be significantly improved, and perhaps slimmed down with some results presented in appendices or combined.
- **Mixed messages**: The methods section sets up the proposed method in competition to alternative methods (such as probcover and typi). This is most prominent in the description of computational complexity, where the proposed method is compared to probcover in complexity. However, ultimately the methods are all combined with one another in the results sections. This leads to a disconcerting switch between the methods and results sections. I recommend a firming up of this narrative.
- **Lack of discussion section**: The conclusion of this work appears rather suddenly after a barrage of results. This leads to very little idea of what the implications and real take-aways of the results are. A discussion section which brings together the set of findings into a set of takeaways could help a great deal in this respect

---

> ### Author Response · Authors · 2024-12-12
>
> We thank the reviewer for the helpful comments. We submitted a revised version with colored changes. Here are the changes made following your comments:
>
> 1. The results section has been reorganized for better narrative flow. Key results are now presented together with their corresponding text, and a discussion section has been added. Results with an initial pool were moved to the appendices. In addition, we include a simple example of EigenFaces which supports our message and the motivation of our ideas.
>
> 2. We acknowledge the concern about our method performance and we now clearly represent the impact of our method, including its limitations. Additionally, we include additional results of fine-tuning of BERT with Yelp and enhance the model-agnostic quality of our method with results with MobileNet. Additionally, we present the average performance across multiple subset sizes for all tested datasets in Table 2. The results clearly demonstrate that, on average, incorporating the norm criterion is highly beneficial.
>
> 3. The methods section now includes a detailed explanation of how our proposed approaches (Norm and Gram-Schmidt) are integrated with existing methods like TypiClust and ProbCover. We clarify that these combinations are flexible and can be extended to other subset selection methods.
>
> We hope these changes address the reviewer’s concerns.

---

### Review · Reviewer_y5ns · 2024-12-02

**Summary Of Contributions:**

This paper introduces the problem of data subset selection (as termed in the paper) from a large, unlabelled pool of samples, which can be used to gather annotations and train a model. This problem is adjacent to the problem of active learning (AL) - unlike AL, this problem selects all of the samples within the subset prior to training. The paper proposes that data subset selection is similar to network pruning if the unlabelled dataset can be considered as the zeroth layer of the model and methods used for network pruning can be applied for selecting the most important samples. Thus, the paper proposes the use of feature norm, which is a method widely used for network pruning, to select data samples. The algorithm proposed in this paper assigns a probability to each data sample, which is computed as the weighted norm, and draws subsets from this distribution. Additionally, the algorithm employs the Gram-Schmidt (GS) process to ensure diversity in the selected subset. Results are shown for 4-5 image classification datasets in three settings i.e., fully-supervised, semi-supervised with linear probing and semi-supervised. Baselines are two strong active learning methods, TypiClust and ProbCover. Experiments are conducted for random sampling, norm-based sampling, norm + GS and norm/GS on top of SoTA baselines. Results show that norm-based sampling generally performs significantly better than random sampling. GS improves over norm-based sampling but it not as good as the SoTA baselines. However, norm and GS do provide additional gains when used with SoTA baselines in most settings.

**Audience:**

Yes

**Claims And Evidence:**

No

**Requested Changes:**

*Critical for Recommendation*
- **Motivation for Subset Selection Task**: I suggest that the authors provide a more extensive discussion as to why the subset selection task as proposed in the paper is important when active learning approaches exists and are widely studied in the community.
- **Empirical Validation for Assumptions**: Section 4.2 needs to be adjusted to provide sound empirical validation for the assumptions and motivations claimed in the paper.
- **Fine-tuning Experiments**: As outlined under Weaknesses, this method can be more impactful in the finetuning regime, so if the authors could include some experiments along those lines, it would support the paper.
- **Model-Agnostic Subsets**: As far as I understand, the model-agnostic quality of the subsets selected in this method is the one advantage over active learning. This claim should be supported with experiments. Training on multiple models for any one of the three settings considered in this paper should suffice.


**Other Minor changes in the paper**:
- Changes to title to clarify *unlabelled data* subset selection.
- Third paragraph in Related work contains repeated sentences.
- Figure 1 is unclear as to which parts are being filtered. Please provide explanation or improve the clarity of the figure.
- Distinction between DINO and SimCLR in 7th paragraph in Intro: Both DINO and SimCLR randomly initialize their models (both teacher and student in DINO), so the distinction presented in the paper i.e., * randomly initialized networks vs. self-supervised networks* is misleading. The distinction lies in their method of training, which is contrastive vs. self-supervised.
- In-line citations: Please use the correct latex command for in-line citations within round brackets.
- Last line in Section 2: .. norm and randomization to select a subset together with an algorithm to ensure feature diversity.
- Appendix & Broader Impact Statement belong in the main pdf.
- Page 4, Line 1: norm of the features
- Complexity: In the complexity discussion in Page 4, I think it should be clarified that the complexity of the algorithm is $O(sNd)$ only when the selected subset is very small i.e., $s << N$. In other cases, the complexity of the algorithm proposed in the paper is $O(N^2d)$.
- Experiments: Add paragraph labels to the section, to guide the reader better e.g., Feature Source, Training Model, Subset Size etc. Please add a corresponding Table to Appendix on experimental settings, since there's a lot of moving parts in the experiments.
- Please try to bring figures to the same page as the part in Results where they are discussed.
- Results from Gram-Schmidt are missing in Figure 2 c,d?
- Figure 4, 6 captions do not mention the experimental setting for the results.

**Strengths And Weaknesses:**

*Strengths*:
- **Subset Selection and Network Pruning**: This paper’s treatment of the data subset selection problem as an extension of the neural network pruning problem is a clever reframing of the subset selection task and opens opportunities for exploring more methods from the domain of network pruning for subset selection.
- **Experiments**: The paper conducts extensive experiments for image classification datasets and in three different settings.
- **Promising Results**: Results show that norm-based sampling is a simple technique that can be used to push the gains from feature-based selection methods. Since it operates on pre-extracted features, it does not significantly impact the computational complexity of the feature-based selection methods either, which is a bonus.


*Weaknesses*:
- **Problem Setting**: The subset selection problem tackled in this paper *is closely related to active learning** but it different in that all of the subset's samples are selected beforehand. The paper hasn't made clear why this is an important problem when active learning is already a widely studied and practical approach to annotating the most important samples. In fact, the baselines used in this paper are also borrowed from active learning papers, which suggests that this not a problem that has been studied in previous works. It is mentioned in passing in the Related Work section that subsets selected in this manner are model-agnostic unlike active learning, but there are no experimental results to support this hypothesis. Further, **subset selection** is a far too broad term to be used in this paper. I suggest that the title of the paper be a bit more explanatory i.e., Unlabelled Data Subset Selection.
- **Experiments in the NLP and Finetuning Regime**: Annotations are frequently needed for finetuning pretrained models on specific tasks, therefore, it makes sense to test such subset selection approached in the finetuning regime. Good subset selection techniques are far more impactful when they can be applied for smaller finetuning datasets, as well as for domains that include language (NLP, multimodal). I acknowledge that it is hard to conduct comprehensive experiments convering all modalities in a single paper, I believe it is easier to run finetuning experiments than pretraining experiments and strongly recommend that authors consider it for the paper.
- **Motivating Hypothesis**: The paper provides some supporting experiments for motivating the case of using high norm data subsets for training (Fig. 7, Table 2). At the same time, it also demonstrates that a simple max of feature norm results in poor performance (Fig. 6c). Considering the latter, I do not expect high correlation between feature norm and accuracy, as shown by the weak correlation numbers in Figure 7. Further, when the subset is being selected via the Gram-Schmidt process, the subset prioritizes coverage of the distribution rather than maintaining the local density of the distribution, which is not captured with the FID score, as shown by the marginal improvements in FID score numbers in Table 2. Therefore, the motivation presented by the paper for using feature norm falls flat in Section 4.2
- **Writing**: The paper could be improved in terms of writing and organization to facilitate easier reading.

---

> ### Author Response · Authors · 2024-12-12
>
> We sincerely thank the reviewer for their thoughtful and detailed feedback. We have carefully revised the manuscript to address your comments and suggestions. Below, we summarize the changes made in response, with all modifications highlighted in blue in the revised version.
>
> 1. We have expanded the discussion on the subset selection task in the related work section, highlighting its relevance and divergence from active learning. Specifically, we emphasize the model-agnostic nature of subset selection, which does not require  iterative interactions with the model.
>
> 2. We include a simple example of our method with eigenfaces which uses a linear feature extractor for face recognition. The example supports the motivation and narrative of our approaches. As for section 4.2, while the correlation with norm is low, it is consistent across settings and serves as a reliable heuristic (when used with randomization). We also corrected the placement of the FID metric to the results subsection since it can serve as another measure of the effectiveness of the selected subsets.
>
> 3. We included experiments fine-tuning BERT on the Yelp dataset, demonstrating that our approach usually improves performance.
>
> 4. We present the average performance across multiple subset sizes for all tested datasets in Table 2. The results clearly demonstrate that, on average, incorporating the norm criterion is highly beneficial.
>
> 5. We now include MobileNet fine-tuning results in the appendix, supporting the model-agnostic nature of our approach. Additionally, we emphasize that our evaluations inherently cover different models, including ResNet, linear classifiers, and semi-supervised WideResNet, further validating this claim.
>
> Minor Change:
>
> 6. The title has been updated to "Effective Unsupervised Subset Selection…”
>
> 7. DINO and SimCLR are indeed intialized randomly but we meant for trained DINO and SimCLR and randomly initialize model without training. This is clarified in the revised version.
>
> 8. We clarified figure captions, ensured alignment between figures and the text discussing them, and added a table in the appendix detailing experimental settings. Results missing in Figure 2c. have been added. We do not include results for GramSchmidt due to limited resources (this is true for figure 3 too). Although the appendices contain extensive results, we believe that maintaining conciseness in the main body of the paper is essential; therefore, we have opted not to incorporate the appendices into the main text.
>
> 9. The complexity discussion has been refined to specify that the algorithm's complexity is O(sNd) for small subsets (s≪Ns) and O(N^2d) otherwise.
>
> We hope these changes address the reviewer’s concerns.

---

### Author Response · Authors · 2024-12-23
**Gentle Reminder to Review the Revised Manuscript**

We are writing to kindly remind you to review the revised manuscript and our responses to your insightful comments and suggestions. Your feedback has been invaluable in refining the work, and we have made efforts to address all points raised, providing clarifications and incorporating changes where necessary.

We would greatly appreciate it if you could take some time to review the revision and let us know if there are any further concerns, questions or areas that require additional clarifications.

Thank you for your support, time and efforts you dedicate to this review process.

---

### Decision · Action_Editor_kLmy · 2025-01-29

**Recommendation:** Reject

**Comment:**

This paper applies principles from neural network pruning methods for the purpose of data sample selection for annotation. The selection of data for annotation is an important problem for a number of domains in which annotation is expensive and data is limited in scale. This work proposes the perspective of data samples as if they were sub-components of a neural network and, as such, could be given an importance score and pruned (or retained) based upon this score. Borrowed from network pruning, this work proposes the measurement of the L2 norm of feature (data) samples and a re-weighting of the selection of these samples based upon this norm. Furthermore, they propose the addition of an orthogonalizing process to reduce the weighting of samples based upon their similarity to existing selected samples. This allows the selection of an entire subset of data samples, for the purpose of annotation and training. The impact of the addition of such components to the sample-selection pipeline during training neural networks is explored in the results section.

However, there are several aspects to be further improved. For example, this paper does not substantiate its claims convincingly due to the following issues: The proposed method and evaluation scheme appear fundamentally flawed. The authors leverage more powerful pretrained backbones for feature extraction and subsequently train the same or smaller model on a dataset like CIFAR-10. This approach does not present a clear advantage, as the pretrained model is already accessible. This raises a significant concern: the proposed method seems limited to scenarios where the dataset is within the feature extractor's distribution. If this is the case, it undermines the necessity of training a new model entirely. One can zero-shot the pre-trained model or simply fine-tune it. Conversely, if the dataset is out-of-distribution (OOD) for the feature extractor, I am skeptical about whether the extracted features (e.g., norms) would remain meaningful or indicative. Even after considering the rebuttal, the model backbones and datasets used in the experiments are overly simplistic. For the proposed method to demonstrate its broader applicability and effectiveness, experiments on more robust and contemporary backbones—such as models in the scale of LLaMA3-8B—are necessary. Without this, the method's claims remain unconvincing for real-world scenarios. The authors mention that the feature extractor may require unsupervised training, introducing additional computational costs. This further accentuates the need to benchmark the method against existing literature on training data attribution, which has been extensively studied and shown to be highly effective. The lack of such comparisons weakens the paper's contributions and contextual significance. Given the above limitations, the proposed method may need to be revised accordingly. The authors may consider submitting a major revision at a later time.

**Audience:**

Yes, this paper applies principles from neural network pruning methods for the purpose of data sample selection for annotation.

**Claims And Evidence:**

No, this paper does not substantiate its claims convincingly due to the issues in comments.

**Resubmission Of Major Revision:**

The authors may consider submitting a major revision at a later time.

---

> ### Author Response · Authors · 2025-02-11
> **A mistake in the editor comment**
>
> We thank the editor and reviewers.
> Yet, there is a major mistake in the meta review of the editor.
> In our experiments, the pre-trained networks that we use were pre-trained in a self-supervised way and therefore all the suggestions of zero shot or fine-tuning are non-relevant. Note also that we show results with networks that are randomly initialized.
>
> Regarding the setup, we follow a previous work and therefore we selected the same datasets they used. We also added a new language dataset and showed improvement also on it. While it is possible to run on larger datasets, we dont have the resources for it and given that prior works demonstrate their work on the same setup, we believe that it is an unfair request.
>
> Given that the editor missed the fact that the Pretraining was done in a self-supervised way and that we use the same datasets like prior works (with language addition), we kindly request the editor to re-consider the decision